# Using passive and active observations at microwave and sub-millimetre wavelengths to constrain ice particle models

Robin Ekelund[1], Patrick Eriksson[1], and Simon Pfreundschuh[1]

[1]Department of Space, Earth and Environment, Chalmers University of Technology, Gothenburg, Sweden

**Correspondence:** Robin Ekelund (robin.ekelund@chalmers.se)

**Abstract.** Satellite microwave remote sensing is an important tool for determining the distribution of atmospheric ice globally. The upcoming Ice Cloud Imager (ICI) sensor will provide unprecedented measurements at sub-millimetre frequencies, employing channels up to 664 GHz. However, the utilization of such measurements requires detailed data on how individual ice particles scatter and absorb radiation, i.e., single scattering data. Several single scattering databases are currently available, with the one by Eriksson et al. (2018) specifically tailored to ICI. This study attempts to validate and constrain the large set of particle models available in this database, to a smaller and more manageable set. A combined active and passive model framework is developed and employed, which converts CloudSat observations to simulated brightness temperatures (TBs) measured by the GPM Microwave Imager (GMI) and ICI. Simulations covering about one month in the tropic Pacific ocean are performed, assuming different microphysical settings realized as combinations of particle model and particle size distribution (PSD).

Firstly, it is found that when the CloudSat inversions and the passive forward model are considered separately, assumed particle model and PSD have a considerable impact on both radar retrieved ice water content (IWC) and simulated TBs. Conversely, when the combined active and passive framework is employed instead, the uncertainty due to assumed particle model is significantly reduced. Furthermore, simulated TBs for almost all the tested microphysical combinations, from a statistical point of view, agree well with GMI measurements (166, 186.31, and 190.31 GHz), indicating the robustness of the simulations. However, it is difficult to identify a particle model that outperforms any other. One aggregate particle model, composed of columns, yields marginally better agreement to GMI compared to the other particles, mainly for the most severe cases of deep convection. Of tested PSDs, the one by McFarquhar and Heymsfield (1997) is found to give the best overall agreement to GMI and also yields radar dBZ-IWC relationships closely matching measurements by Protat et al. (2016). Only one particle, modelled as an air-ice mixture spheroid, performs poorly overall. On the other hand, simulations at the higher ICI frequencies (328.65, 334.65, and 668.2 GHz) show significantly higher sensitivity to the assumed particle model. This study thus points to the potential use of combined ICI and 94 GHz radar measurements to constrain ice hydrometeor properties in radiative transfer (RT), using the method demonstrated in this paper.

## 1 Introduction

Active and passive microwave instruments are useful for measuring the mass of ice hydrometeors. This is for example relevant for deep convection systems and associated anvil cirrus (high altitude ice clouds), which are dominant features of the intertrop-

ical convergence zone and play a major role in the earth radiative energy budget, climate, and hydrological cycle. Accurate measurements of these tropical clouds are important from a weather and climate prediction point of view. Because clouds are semi-transparent at microwave frequencies, measured signals are related to the integrated column of ice hydrometeors. Depending on the exact measurement frequency, sensitivity to different particle sizes is gained. Radars utilize frequencies up to 95 GHz and are predominantly sensitive to larger, presumably precipitating hydrometeors, e.g., snow. Radiometers that make use of frequencies above 300 GHz, i.e., sub-millimetre frequencies, are instead sensitive to smaller particles, useful for measuring ice clouds (Buehler et al., 2007). In order to estimate the total ice mass accurately, a suitable combination of frequencies should be chosen in order to offer sensitivity to the particle sizes contributing the most to the total mass for varying conditions.

In order to provide global datasets, satellite-borne instrumentation is required. Microwave radiometers typically provide excellent horizontal coverage, but poor vertical resolution (Bühl et al., 2017). As a prominent example, the Global Precipitation Mission (GPM) core satellite carries the GPM Microwave Imager (GMI), a conical scanning microwave radiometer with channels up to 183.31 GHz, for estimating rain and snowfall (Hou et al., 2014). Overall, however, there is currently a lack of global data on ice cloud mass, since present sensors operate at frequencies below 190 GHz, where the sensitivity to smaller particles is low. The Ice Cloud Imager (ICI) instrument aboard the upcoming MetOp-SG satellite will address this issue, employing channels in the sub-millimetre region, at frequencies up to 664 GHz. The suitability of sub-millimetre wavelengths for retrievals of cloud ice has been demonstrated practically by a number of airborne instruments, including the airborne ICI demonstrator International Sub-millimetre Airborne Radiometer (ISMAR) (Fox et al., 2019; Brath et al., 2018) and the Compact Scanning Submillimeter-wave Imaging Radiometer (CoSSIR)(Evans et al., 2012), but also by a number of limb sounding instruments, including the Aura Microwave Limb Sounder (MLS) (Wu et al., 2006), Superconducting Submillimeter Limb Emission Sounder (SMILES), and Odin sub-millimetre radiometer (SMR) (Eriksson et al., 2014).

Satellite-bourne radars, on the other hand, provide excellent vertical resolution globally, at the expense of horizontal coverage (Bühl et al., 2017). Currently operating satellite radars are the GPM dual-frequency phased-array precipitation radar (DPR) operating at 13.6 and 35.5 GHz (Hou et al., 2014) and the CloudSat Cloud Profiling Radar (CPR) operating at 94.1 GHz (Stephens et al., 2008). On ground, weather radars are typically used to estimate precipitation, but are also extensively used for research purposes. Furthermore, Doppler radars have demonstrated capabilities of both identifying and quantifying different water phases in mixed-phase cloud (Shupe et al., 2004), while multi-frequency radars have shown capabilities to retrieve information related to the shape of the ice particles (Kulie et al., 2014; Kneifel et al., 2015; Yin et al., 2017; Leinonen et al., 2018).

In order to utilize remote sensing measurement affected by ice hydrometeors, the single scattering properties of the ice particles must be known. Developing particle models that produce realistic scattering over a large frequency spectrum is difficult. Traditionally, simplistic particle models such as the soft spheroid particle (i.e., a spheroid composed of an ice/air mixture) has been used both in the active and passive remote sensing communities, due to its computational efficiency. Such configurations can yield good results at single frequencies, for instance, through fine-tuning of the particle effective density (Galligani et al., 2015). However, they fail at providing consistent results at multiple frequencies and are not appropriate for multi-frequency measurements and combined passive/active applications (Geer and Baordo, 2014).

These issues have triggered the development of more advanced representations of ice particles in radiative transfer (RT), often with either passive or active applications in mind. Scattering calculations of complex particles are computationally demanding; hence, databases of pre-calculated single scattering properties are a necessity. The work of Liu (2008) and Hong et al. (2009) could be considered the first efforts at such single scattering databases, aimed specifically at microwave applications. More recent attempts at single scattering databases were published by Lu et al. (2016); Kuo et al. (2016); Ding et al. (2017); Eriksson et al. (2018). The database by Eriksson et al. (2018), coined the ARTS (Atmospheric Radiative Transfer Simulator) database, was mainly developed in support of the upcoming ICI mission, requiring a frequency coverage up to 668 GHz. A wide range of different particle models is included, with an emphasis on realistic aggregates. However, the wide range of particle models presents a problem and the following question: Which models are the best representations of ice hydrometeors in reality?

This study is an attempt to assist the utilization of future ICI observations. In specific, the aim is to validate a selection of particle models of the ARTS database and, if possible, also constrain the number of particle models to a smaller, more manageable set. The validity of a particle model is in this context its ability to accurately reproduce sensor observations in RT modelling. The ARTS database has already seen some analysis in the study by Fox et al. (2019), which reproduced airborne passive measurements by ISMAR. The simulations were based on measurements by lidar and in-situ probes that were also carried by the aircraft. It was found that measurements could be simulated with good accuracy. However, accuracy depended heavily upon assumed particle model and which part of the cloud (e.g., altitude, thickness, age, etc.) that was considered, implying that accurate ICI retrievals of cloud ice mass will require realistic particle models. However, the study, while detailed, was limited to two cases of thin cirrus clouds at mid-latitudes. It is of interest to perform a more statistically robust evaluation of particle models, i.e., in a context more globally representative.

In general, validation of particle models is difficult. In-situ aircraft and ground-based measurements give detailed information on particle shape and size distribution, but the coverage is generally highly localized. For mesoscale and larger systems, forward simulations of satellite observations can be used to investigate the impact of the microphysical assumptions upon remote sensing measurements. Typically, atmospheric fields and surface properties taken from a cloud resolving model are used as input to the forward calculations (Meirold-Mautner et al., 2007; Hong et al., 2010). Furthermore, if real observations are available, they can be confronted with simulated observations to constrain microphysical parameterizations (Sreerekha et al., 2008; Geer and Baordo, 2014; Galligani et al., 2015, 2017). For instance, Geer and Baordo (2014) evaluated the particle models of the database by Liu (2008) by comparing observed (by TMI and SSMIS) to simulated brightness temperatures (TBs) using ECMWF (European Centre for Medium-Range Weather Forecasts) forecasts as input. They found that agreement was improved for most of the Liu particle models, in comparison to Mie spheres. However, using model input presents a number of problems. The RT model must be realistic and if the spatial resolution of the model is too coarse, local high concentrations of ice will not be resolved (Burns et al., 1997). Also, numerical weather prediction can typically not predict small-scale cloud structures, making comparisons between simulated and real observations difficult at such scales.

This issue can be circumvented by using radar reflectivity fields as input to the forward model (Bennartz and Bauer, 2003; Skofronick-Jackson et al., 2008; Kulie et al., 2010; Yin and Liu, 2019; Ringerud et al., 2019), ensuring that small-scale cloud

structures are properly resolved and represented. Lidar measurements can be used in a similar way, as the already mentioned study by Fox et al. (2019) is an example of. However, this approach is limited to thin clouds that lidars can penetrate. The utilization of measurements by radar requires that the reflectivity fields are converted to fields of ice water content (IWC) (i.e., mass density of ice hydrometeors). The recent study by Yin and Liu (2019) assessed GMI measurements by comparing them to forward simulations derived from collocated CloudSat measurements, using a selection of different non-spheroidal particle models. In general, good agreement was attained, especially for the aggregate particle type used. However, Yin and Liu (2019) only derive the IWC field once and used it as input to the forward simulations regardless of the particle type assumed in the latter procedure. This implies that the microphysical assumptions in the forward simulations and the active retrievals are not necessarily consistent. Kulie et al. (2010) assessed particle models by performing RT simulations mimicking passive sensors (AMSR-E and MHS), using IWC input derived from collocated CloudSat CPR measurements as well. Here, the IWC fields are derived repeatedly, with consistent microphysical assumptions in the forward simulations. Hence, consistency between the passive and active model was established. They found that that the simplistic sphere particle models could not produce results consistent with both the CPR and passive measurements. However, using more complex particle models, extracted from scattering databases by Liu (2008) and Hong et al. (2009) performed better overall. No particle was found to perform well in all conditions (e.g., cold clouds, warm clouds, stratiform, etc.), however.

Following the work by Kulie et al. (2010), an effort is made to evaluate the performance of a set different particle models from the ARTS database. Multiple parametrizations of particle size distributions (PSDs) from different sources are considered, since the assumed PSD has a large impact upon simulated observations of clouds. A modelling system that takes radar measurements as input and simulates passive TB measurements was developed. The model operates in two modes. In the first mode, IWC fields from the DARDAR cloud product are used as input. In the second mode, CloudSat radar reflectivities are used as input instead, which are converted to IWC fields using particle model and size distributions identical to those assumed in the consequent passive forward simulations. Retrieved IWC fields are combined with atmospheric data from the ERA-interim reanalysis in order to generate synthetic scenes as input for the forward simulations. TBs are simulated at the highest frequencies of GMI using different combinations of particle model and PSD. The simulated measurements are then compared to GMI observations in a statistical sense, in order to test the different microphysical combinations. The scope is limited to daytime simulations in the tropical Pacific ocean, over a period of roughly one month. The geographic location avoids issues associated with simulating over land and also ensures that a wide range of weather conditions are covered, including severe storms. Finally, simulations are also performed at the highest ICI frequencies, to indicate the validity and possible added value of adding ICI measurements to the analysis performed in this paper.

The paper is structured as follows: Section 2 overviews used software, satellite data, and microphysical data. Section 3 outlines the overall simulation setup and settings, and also describes the generation of synthetic scenes. Section 4 presents the results. Finally, discussion, conclusions, and outlook are found in Sect. 5 and 6.

## 2 Data and simulation tools

### 2.1 CloudSat

CloudSat, a sun-synchronous satellite launched in 2006, carries a 94.05 GHz, nadir-looking cloud profiling radar (CPR) (Stephens et al., 2008). The range resolution is 485 m, while the along- and cross-track resolution is 1.8 and 1.4 km, respectively. As a part of the A-train constellation, the CloudSat CPR has provided collocated products with the Cloud-Aerosol Lidar with Orthogonal Polarization (CALIOP) aboard the CALIPSO satellite. DARDAR (Delanoë and Hogan, 2010), is a synergetic ice cloud retrieval product combining measurements by these two sensors, and is one of the most advanced dataset

of clouds to date. Bear in mind that radar attenuation by ice and liquid is neglected in this product, however.

The CloudSat data used in this study are extracted from the DARDAR product. Parameters used are DARDAR IWC, DARDAR normalized number concentration parameter ($N_0^*$) and CloudSat radar reflectivity. The time frame of selected data is from 3 July to 1 August 2015. Geographically, data are extracted from a rectangular area in the Pacific ocean, placed directly at the equator. The area is limited in latitude by S20° and N20°, and in longitude by E170° and W130°. The reason for the

140 selected spatial and temporal frame is given in Sect. 3.1.

### 2.2 GMI

The Global Precipitation Mission (GPM) Microwave imager (GMI) is a conically scanning radiometer with 13 channels in total (Hou et al., 2014). The frequencies of this sensor range from 10.65 to 183.31 GHz, with both h-and v-polarization at certain frequencies. The channels from 89 to 183.31 GHz have footprint sizes of roughly 6×6 km. GMI is mounted on the

145 GPM satellite, which serves as the core satellite of the GPM constellation. The satellite has a non-sun-synchronous orbit with a 65° inclination, which allows for collocated measurements with other satellites in the GPM constellation. GMI can, therefore, provide inter-sensor calibration for the GPM constellation satellites, for the purpose of improved consistency among precipitation products. This study makes use of the vertically polarized measurements at 186.31 and 191.31 GHz. Data are selected within the same spatial and temporal frame as for the DARDAR data, as described in Section 2.1.

### 150 2.3 ICI

The Ice Cloud Imager (ICI) is an upcoming conically scanning radiometer that will fly on the second generation meteorological operational satellite (MetOp-SG B), operated by the European Organisation for the Exploitation of Meteorological Satellites (EUMETSAT) and scheduled for launch in 2023. Its main goal is to provide global measurements of ice hydrometeors and it will fly in a polar sun-synchronous orbit. ICI has 13 channels from 183.31 to 664 GHz. Channels placed at 243 and 664 GHz

measure both vertical and horizontal polarization, while the other channels measure vertical polarization only. ICI scans with a 53° incidence angle and has a footprint of 15 km. Forward simulations were performed for ICI channels at 328.65, 334.65, and 668.2 GHz.

## 2.4 ERA-Interim

ERA-interim is a global atmospheric reanalysis provided by ECMWF. The parameters used here are pressure, surface, atmospheric temperature, humidity, and cloud liquid water content (CWC). The humidity and CWC fields are not used without modification; they are adjusted in order to be more realistic and agreeable with merged CloudSat data. These modifications are described in detail in Sect. 3.1.

## 2.5 ARTS

The Atmospheric Radiative Transfer Simulator (ARTS) is an open-source software package that focuses on simulating longwave RT (Buehler et al., 2018). It is intended to describe radiation using the full Stokes vector notation in the most general manner possible, allowing a large amount of user input flexibility. In itself ARTS behaves as a scripting language on its own. It is oriented towards accuracy and treats particle scattering in a rigorous manner, making it ideal for usage in this study. Several scattering algorithms are available and have been used, including Monte-Carlo (Rydberg et al., 2009), RT4 (Fox et al., 2019) and DOIT (Brath et al., 2018). Also available is the DISORT (Discrete Ordinates Radiative Transfer Program for a Multi-Layered Plane-Parallel Medium) algorithm, which is limited to unpolarized radiation. This study only considers totally randomly oriented scattering particles, so this limitation is not a major concern. Furthermore, the method is easy to use, insensitive to settings and stable. Hence, DISORT was selected as the scattering solver to be used in this study.

## 2.6 Single scattering data

The ice particle single scattering properties to be evaluated in this study are provided by the publicly available database by Eriksson et al. (2018), referred to as the "ARTS scattering database", as it covers the scope of this study in terms of frequencies and selection of particle models. It provides scattering data for a total of 34 particle models, 34 frequencies between 1 to 886.4 GHz, and three temperatures at 190, 230 and 270 K.

The particle models, or habits as they are also commonly called, selected for evaluation in this study are outlined in Tab. 1. A fundamental difference between the particle types is how the particle mass is related to the size as they grow. Traditionally, this relationship is described by a power-law

$$m = aD_{\mathrm{max}}^b, \tag{1}$$

where $m$ is the particle mass, $D_{\mathrm{max}}$ the maximum diameter, and $a$ and $b$ the coefficients that are specific to the particle model. Table 1 lists values of $a$ and $b$ for each particle model, calculated by fitting a line to the mass-size relationship in logarithmic space; instances of $D_{\mathrm{max}}$ below 200 μm are not included the calculations. Here, $D_{\mathrm{max}}$ is defined as the diameter of the minimum circumsphere of the particle. Because some of the particle types do not cover broad enough size spans, there are also particle mixes available. Particle mixtures consist of a pristine crystals at lower sizes and more complex particles like aggregates at larger sizes. There is a gradual shift in particle and scattering properties in a transition region, in order to avoid discontinuities. The width and placement of the transition region differs between particle models, but is roughly between 100

**Table 1.** Summary of particle models used. Given sizes are in mm and refer to largest particle size of each model. Further details are found in the text.

| Particle model | $D_{\text{veq}}$ | $D_{\text{max}}$ | a | b |
|---|---|---|---|---|
| Evans snow aggregate | 2.51 | 11.8 | $3.1 \cdot 10^{-2}$ | 2.07 |
| Sector snowflake | 1.31 | 10.2 | $7.2 \cdot 10^{-3}$ | 1.79 |
| 8-column aggregate | 5.00 | 9.7 | $6.5 \cdot 10^{1}$ | 3.00 |
| ICON cloud ice | 2.92 | 10.0 | $1.6 \cdot 10^{0}$ | 2.56 |
| Large plate aggregate mix | 4.56 | 22.9 | $5.1 \cdot 10^{-1}$ | 2.41 |
| Large column aggregate mix | 3.02 | 20.0 | $3.8 \cdot 10^{-1}$ | 2.50 |
| Large block aggregates mix | 4.61 | 21.9 | $7.2 \cdot 10^{-1}$ | 2.39 |
| ICON snow mix | 3.22 | 20.0 | $2.1 \cdot 10^{-1}$ | 2.79 |
| DARDAR spheroid | 2.00 | 14.3 | $1.3 \cdot 10^{-2}$ | 1.91 |
| Liquid spheres | 50.0 | 50.0 | $5.2 \cdot 10^{2}$ | 3.00 |

to 400 μm. Since the reported $a$-and $b$-values are calculated for particle mixtures, they differ somewhat to values reported in Eriksson et al. (2018). The database also includes scattering data for liquid spheres, calculated using Mie code, which will be used to describe scattering by rain.

A soft-spheroid particle is also included, whose scattering properties are calculated using T-matrix code by Mishchenko (2000). The particle is designed to mimic the settings applied in the DARDAR v2.1 product. In short, the spheroids are oblate with an aspect ratio (ratio of the minor to the major axis) of 0.6, and follow the mass-size relationship given in Delanoë et al. (2014, Eq. 13-15). The refractive index of the particles are calculated using the Maxwell-Garnett mixing formula (Garnett, 1904) assuming an air-in-ice mixture. Eriksson et al. (2015) evaluated mixing rules in the context of soft spheroid approximations and found that this mixing formula performs relatively good compared to DDA data.

While $D_{\text{max}}$ is commonly used to characterize the size of particles, the volume-equivalent diameter $D_{\text{veq}}$ will be used throughout in this study. It is equivalent to the particle mass, which is more related to IWC and is generally more connected to the microwave particle scattering properties than $D_{\text{max}}$ (Ekelund and Eriksson, 2019).

In order to assist the evaluation of the impact of assumed particle type on resulting simulations presented later, an overview of the scattering properties of the included particle types is presented in Fig. 1, displaying the back-scattering at 94.1 GHz and extinction at 190.31 and 668.2 GHz, as a function of $D_{\text{veq}}$ for all used particle types, in the left, middle and right panel, respectively. Back-scattering and extinction are given as cross-section ratios in decibels, using the 8-column aggregate as the reference. The 8-column aggregate therefore shows up as a horizontal line in all of the panels.

The ICON cloud ice, 8-column aggregate, and sector snowflake are found to be relatively efficient back scatterers, while the Evans snow aggregate, large column aggregate, and especially the DARDAR-spheroid are relatively inefficient. The spread in

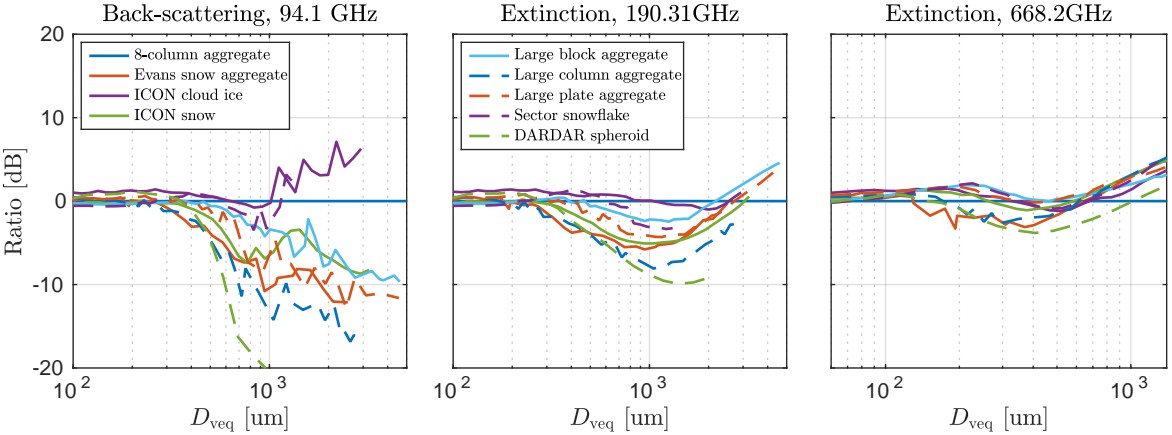

**Figure 1.** Back-scattering ratio at 94.1 GHz (left) and extinction ratio at 190.31 GHz (middle) and 668.2 GHz (right) as a function of volume equivalent diameter $D_{veq}$. Legends are common to all panels. The ratios are calculated as the cross-section of given particle model divided by the cross-section of the 8-column aggregate and are given in dB. That is, -3 dB means that the back-scattering or extinction cross-section is half of the Hong aggregate cross-section. All data assume totally random particle orientation.

back-scattering is also enhanced as particle size increases. In the middle panel, the ordering of particle models in extinction is similar. For instance, the 8-column aggregate is a very efficient scatterer both at 94.1 and 190.31 GHz. However, the spread

between the particles is lower in comparison to the left panel. At sizes above 1 mm the lines converge and the 8-column aggregate is gradually replaced by other particles as more efficient scatterers. At 668.2 GHz, the relative spread between particles is even more compact. Above 500 μm, most of the particles, except the DARDAR spheroid and the 8-column aggregate, have aggregated to a cluster of lines with a spread of approximately 2 dB.

## 2.7 Particle size distributions

The particle size distribution (PSD) describes, as its name suggests, how the particle sizes are distributed within a volume element. By integrating over all sizes, the particle number concentration is attained, while the particle mass density is attained by weighting with the particle mass. IWC is given by

$$\text{IWC} = \int_0^\infty m(D_{veq})N(D_{veq})\,dD_{veq}, \tag{2}$$

where $N$ describes the PSD and $m$ is the mass of individual particles (in this study given by Eq. 1). Three different PSDs

were selected for this study, where two of them are commonly used by the passive community and the last one by the radar community. The first PSD by McFarquhar and Heymsfield (1997), denoted as MH97, has been used predominantly in limb microwave measurement related studies (Davis et al., 2005; Wu et al., 2006; Eriksson et al., 2007; Rydberg et al., 2009). In-situ data were collected from the Central Equatorial Pacific Experiment (CEPEX). It should mainly be considered valid for anvil cirrus in the tropics. Consequently, MH97 places relatively low weight on large particles, and is therefore less suitable for

snow, for instance. In contrast to many other PSD parametrizations, MH97 make use of the volume (mass)-equivalent diameter $D_{\text{veq}}$ as the size descriptor. The MH97 PSD is therefore mass-conservant, i.e., the PSD-integrated mass does not depend on the mass-size relationship of the particle model.

The second PSD used here was developed by Field et al. (2007) and is denoted as F07. F07 and its predecessors have arguably seen more widespread use compared to MH97 (Kulie et al., 2010; Geer and Baordo, 2014; Fox et al., 2019). F07 is

230 a single moment PSD parametrization based on in-situ PSD data compiled from multiple measurement campaigns. As input it requires the temperature and the mass-size relationship (Eq. 1) of the employed particle model, and also has two different settings for mid-latitude and tropical conditions. Here, only the tropical setting used, denoted as F07T. It uses $D_{\text{max}}$ as the size descriptor and is, therefore, not mass-conservant with respect to assumed particle model.

The PSD used by the DARDAR v2.1 product is employed as well, in order to provide insight to this specific product. It

is described in Delanoë et al. (2014) and is referred to as D14. It is a modified gamma distribution fitted to in-situ data. As such, the same $\alpha_{\text{F}}$ and $\beta_{\text{F}}$ coefficients used in DARDAR version 2 are used for this study (Cazenave et al., 2019) ($\alpha_{\text{F}} = -2$ and $\beta_{\text{F}} = 4$). As is the case for MH97, this PSD takes $D_{\text{veq}}$ as input and is mass-conservant. Also, the PSD requires two moments as input, the normalized number concentration parameter $N_0^*$ and the mean volume-weighted diameter $D_{\text{m}}$. Any of these parameters can be interchanged with IWC through conversion, so only IWC and $N_0^*$ are used in this study. Since the

radar inversion algorithm, described later in Sect. 3.2, only retrieves IWC, not $N_0^*$, an a priori parametrization of $N_0^*$ is used instead. The parametrization, described in Delanoë et al. (2014), takes temperature as input and is defined as

$$\ln(N_0^*) = kT + a, \qquad (3)$$

where $k = -0.076586$ and $a = 17.948$ are coefficients fitted to aircraft in-situ measurements and $T$ the temperature given in °C.

245 For describing rain, the PSD by Wang et al. (2016) was selected. It is described by a gamma function, based on aircraft in-situ measurements of multiple convective cores in North Dakota, USA.

The impact of the assumed PSD will be investigated later and an overview of the used PSDs is therefore given in Fig. 2. The panels show the PSD-weighted extinction cross-section of the ICON cloud ice particle, using the PSDs of MH97, F07T, and D14, in the left, middle, and right panel, respectively. Different values of IWC are assumed in the coloured lines, and the

250 distributions have been normalized to have an area equal to $10^{-3}$. Generally, the PSDs produce mostly small particles, with diameters of roughly $10\,\mu\text{m}$. However, Fig. 2 shows that in terms of scattering impact, intermediately sized particles (300 to $1000\,\mu\text{m}$) dominate. Of the investigated PSDs, MH97 puts the highest emphasis on small particles, with two modes being clearly visible for an IWC of $10\,\text{mg}\,\text{m}^{-3}$. There is little change as IWC increases, apart from the first mode becoming less pronounced. F07T and D14 are fairly similar to each other, both resulting in a stronger emphasis on larger particles compared

to MH97 at increasing values of IWC.

As a final comment, the F07 PSD requires the mass-size relationship coefficients as input (a and b in Tab. 1). For comparison, the 8-column aggregate is included in dashed lines in the middle panel. This particle model has a value of b= 3, resulting in

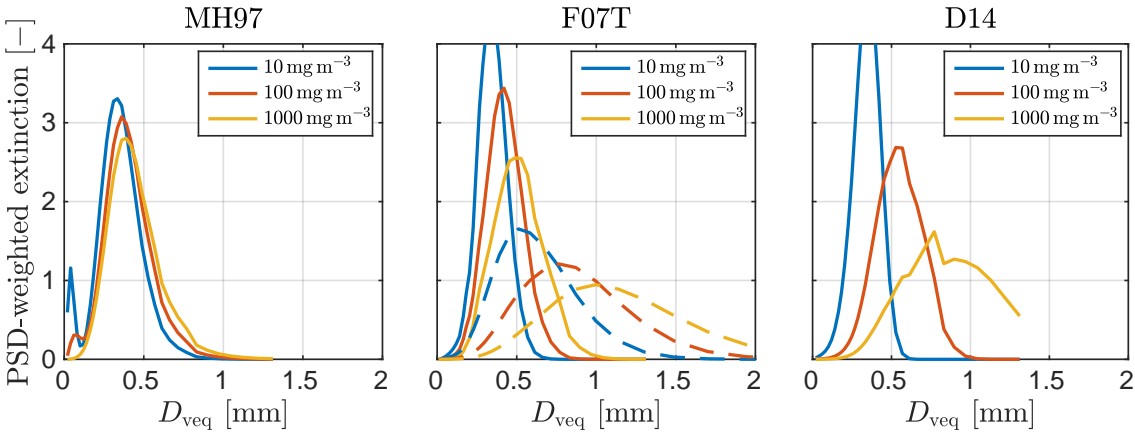

**Figure 2.** Normalized extinction as a function of volume equivalent diameter $D_{\mathrm{veq}}$, for different combinations of particle size distribution and IWC. Frequency is 190.31 GHz, temperature 255 K and assumed particle model is the ICON cloud ice. Each extinction curve is normalized to have an area equal to $10^{-3}$. In the middle panel, extinction calculated using the 8-column aggregate has been included in dashed lines for comparison.

a shift of the distributions towards higher values of $D_{\mathrm{veq}}$. Hence, comparisons between F07 and other PSDs are generally not straight-forward, but depend on assumed particle model.

## 3 Methodology

This section describes the employed modelling system, which simulates passive satellite observations of clouds based on CloudSat radar measurements. There are two variants or modes of this system, denoted IWC-based and dBZ-based, and they are both schematically described in Fig. 3. The overall idea is to perform RT simulations on synthetic scenes, using different microphysical assumptions in order to evaluate their impact on measurements and performance in relation to GMI. These assumptions are realized as different combinations of particle models (9 in total, see Tab. 1) and PSDs (3 in total, see Sect. 2.7). The synthetic scenes are created using data outlined in Sect. 2, and their generation is described in Sect. 3.1. Forward simulations mimicking GMI and ICI are then performed at selected frequencies. These simulations are then compared to GMI measurements statistically. Simulations at ICI frequencies are performed in order to investigate the potential of using ICI in the same way in the future.

The two model modes differ in how the input IWC field is set in the forward model. The first mode (marked blue in Fig. 3) make use of the DARDAR IWC product, invariant to the microphysical assumptions used in the forward model. The purpose of this mode is mainly to indicate the importance of correct microphysical assumptions in situations where IWC is already set or known. The second mode (marked red in Fig. 3) takes CloudSat reflectivities and converts them to IWC fields using an onion-peeling retrieval method (described in Sect. 3.2), with microphysical assumptions consistent to those used in the forward

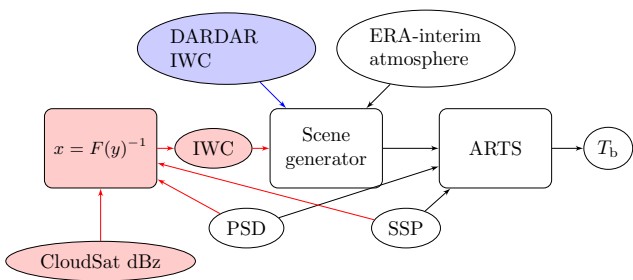

**Figure 3.** Flowchart describing the active and passive modelling system. White boxes and black arrows represent processes and data common to both variants of the system. The blue box and arrow are part of the IWC-based system, i.e., simulated TBs are generated using DARDAR IWC as input. The red boxes and arrows, on the other hand, describe the dBZ-based model system, where the forward simulations use CloudSat reflectivities as input. As indicated by the red arrows from the PSD and SSP (single scattering porperties) boxes, the dBZ-based system assume the same microphysics for both the radar inversion and the consequental forward simulation.

simulation. This mode is used to constrain possible particle and size distribution combinations, forcing them to be consistent for both passive (GMI) and active (CloudSat) observations.

There are a number of limitations to this study that should be noted. Firstly, it is assumed that the orientation of all particles is completely random. The ARTS database currently does not provide scattering data of oriented particles, and there are no appropriate alternative databases that does so. Secondly, for a given simulation, only a single ice particle type is assumed for

the whole scene, i.e., ice particle types for different cloud or precipitation types (e.g cirrus or snow) are the same within a given simulation. Thirdly, comparisons of real and simulated observations using using collocated datasets would have been ideal. However, because the number of collocations between CloudSat and GMI is limited around the equator (Rysman et al., 2018; Yin and Liu, 2019) and the different observation geometries present difficulties, the statistical approach was selected instead.

### 3.1 Generating synthetic scenes

The synthetic scenes are defined on 2-dimensional grids in latitude and pressure, using CloudSat measurements as a reference. Extracted data is restricted to a rectangular geographic area in the Pacific ocean, within S20° to N20° and E170° to W130°. The pressure grid is defined up to $42\,\mathrm{hPa}$, and the latitude grid is resolved in steps of roughly 0.01°. The time frame of the scenes was selected with the purpose of getting the best possible match between CloudSat measurements and GMI measurements. CloudSat passes the equator in the Pacific ocean daytime at roughly 13:30. Since GPM has a non-sun-synchronous orbit, a day

in 2015 where GPM passes the equator roughly the same time could be identified and selected as the reference day. A time frame of 30 days centred around this reference day was then set, from which 59 CloudSat overpasses could be identified within the geographic area and selected as references for the synthetic scenes. From this time frame, all available GMI data within the defined geographical area was extracted for comparison to the forward simulations. While the extracted CloudSat and GMI data are not collocated, they are extracted from the same geographic area, same month and were collected at roughly the same

time of the day (within ±4 hours). As a consequence, a much larger dataset can be considered than if only collocations were to be considered.

As mentioned in previous section, the synthetic scenes are composites of data collected form several sources. Atmospheric fields of temperature, humidity, CWC, surface temperature, and elevation, are collected from ERA-interim (see Sect. 2.4). In the IWC-based mode, IWC fields from DARDAR are employed as an input, while in the dBZ-based mode CloudSat retrievals (described in Sec. 3.2) are set as the input, which are converted to IWC fields using microphysical assumptions consistent with the forward simulations (see Fig. 3). Rain water content (RWC) fields are retrieved from CloudSat reflectivities for both modelling systems, since RWC is not provided by DARDAR. All fields are interpolated onto the aforementioned pressure and latitude grid.

Some modifications to the data provided by ERA-interim are performed. Relative humidity is set to 0.9 at locations with water content above $50\,\mu g\,m^{-3}$, in order to ensure that humidity is not artificially too low or high at locations with clouds or precipitation. Liquid water content is also adjusted. It is set to zero at altitudes higher than $10\,km$ and at temperatures lower than $243\,K$. Furthermore, at pixels where IWC or RWC is zero, CWC is set to zero as well. This ensures that ERA-interim does not place liquid water outside clouds and precipitation detected by CloudSat.

## 3.2 Radar inversions

A brief description of the radar inversions is given here. First of all, for a given particle and PSD combination the effective radar reflectivity $Z_e$ at frequency $\nu$ is given as:

$$Z_e(\nu) = \frac{c^4}{\pi^5 \nu^4 |K|^2} \int_0^\infty \sigma_b(D_{veq}, \nu) N(D_{veq}) \, dD_{veq}, \tag{4}$$

where $\sigma_b$ is the back-scattering cross-section and $K = 0.75$ the dielectric factor; it is set to the value used by CloudSat (Stephens et al., 2008). Following convention, the reflectivity is shown in dBZ, i.e., $10\log_{10}(Z_e)$. For all microphysical combinations, dBZ-values are calculated on an IWC grid and stored in IWC-dBZ tables. At temperatures above $0°$, the same formula is applied assuming rain instead, for which a separate RWC-dBZ table is constructed. Radar inversions are then performed for each microphysical combination. An onion-peeling concept is used, implying that the atmosphere is divided into layers. The IWC-value of the top layer is extracted from the IWC-dBZ table given the measured CloudSat reflectivity at that layer. The two-way radar attenuation of this layer is then calculated, taking the updated IWC (or RWC)-value into account. The CloudSat reflectivity of the layer below is then updated to account for the radar attenuation. The maximum correction in dB is set to 3, in order to account for dBZ-enhancement due to multiple scattering (Battaglia and Simmer, 2008). The IWC and RWC grids are thus calculated iteratively, taking the radar attenuation of above layers into account. Absorption by gases and liquid clouds are considered in the same manner as in the forward simulations, as described in the next section.

### 3.3 ARTS setup

This section describes the forward simulations and the ARTS settings used. The RT simulations are 2-dimensional, but approximated using the independent beam approximation, i.e., 1-dimensional calculations are performed along the line of sight of the simulated sensors. Polarization effects are neglected, since randomly oriented single scattering will not have a strong effect on polarization. Hence, only the first Stokes element is simulated. All simulations are performed at an earth surface incidence angle of $53°$, close to the angles used by ICI and the GMI channels mimicked in this study. The simulated frequencies are 166, 186.31, 190.31, 328.65, 334.65, and 668.2 GHz. As this study is of a demonstrative nature, the lower GMI channels are omitted, since these are less sensitive to cloud ice. We also limit the number of covered ICI channels. The antenna pattern of GMI and ICI are approximated post-simulation, by averaging over multiple pencil beams assuming a Gaussian footprint pattern with a full width at half maximum of 6 (166, 186.31 and 190.31 GHz) and 15 km (328.65, 334.65, and 668.2 GHz), respectively. Oxygen and water vapour absorption are calculated using the PWR-98 model by (Rosenkranz, 1998) and nitrogen absorption using the standard profile by (Liebe et al., 1993). Furthermore, liquid clouds are treated using the absorption model by Ellison (2007) (scattering by liquid clouds is omitted).

## 4 Results

### 4.1 CloudSat inversions

This section gives an overview of the CloudSat radar inversions. The radar-IWC tables are first presented, then retrieved IWP for an example scene is shown. Finally, IWC and IWP across all scenes are overviewed statistically.

#### 4.1.1 IWC - dBZ relationships

Figure 4 overviews calculated relationships between radar reflectivity and IWC, which are used for the CloudSat inversions described in Sect. 3.2. The left panel uses the MH97 PSD, middle panel F07T, and right panel D14. For MH97, the spread of the lines is relatively small, roughly a factor 2, since the PSD puts high emphasis on smaller particles where the spread in backscattering is less significant (see Fig. 1, left panel). Conversely, F07T result in a large spread of the lines, almost a full order of magnitude. In other words, F07T result in a higher uncertainty in IWC for a given dBZ, with respect to assumed particle type. In contrast to MH97 and F07T, D14 display more a gradual increase in spread. At low values of dBZ, the spread is fairly small, while for large values the spread is comparable to that of F07T. As shown in Fig. 2, D14 emphasizes smaller particles at low values of IWC, similar to MH97. At higher IWC values, the PSD behaves more like F07T, explaining the gradual increase in spread.

Also included is the radar reflectivity relationship developed by Protat et al. (2016, Eq. 3-5), which is based on co-located airborne radar and in-situ measurements of deep convective systems in the northern part of Australia. Calculated relationships cover the Protat-line fairly well, though MH97 tends to underestimate dBZ in comparison. However, this is expected; since the relationship by Protat is empirical, it is to some degree influenced by azimuthally oriented particles. Azimuthal alignment

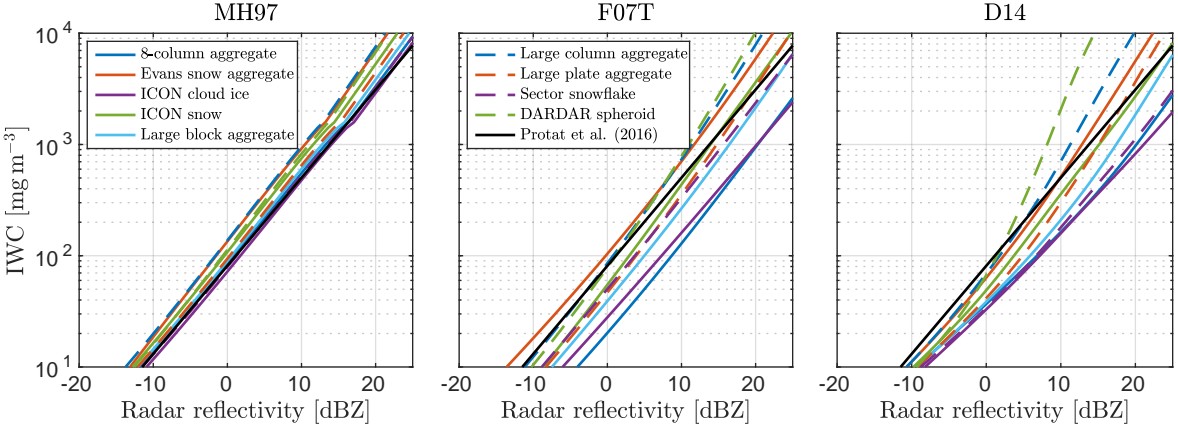

**Figure 4.** Radar reflectivity and IWC relationships at 255 K. The coloured curves are the relationships obtained by combining the particle models of Table 1 with the particle size distributions of MH97 (left), F07T (middle), and D14 (right). The black lines represent the IWC-dBZ relationship taken from Protat et al. (2016).

enhances nadir or zenith backscattering (Hogan et al., 2012), hence, it is expected that the calculated reflectivities are underestimated, by a few dB, in comparison to Protat. Also note that the slopes of the MH97 and Protat radar relationships are almost the same.

### 4.1.2 Example scene

Figure 5 overviews an example scene containing several cloud types, including a deep convective core and some cirrus. The top
panel shows the DARDAR IWC field and the bottom ice water path (IWP), defined as column-integrated IWC. The coloured lines represent IWP from radar inversions as described in Sect. 3.2 (colors are the same as in Fig. 4 or 6) and the black line is IWP derived from the DARDAR product. The spread in IWP between different particles is almost an order of magnitude. The ordering of the lines is explained by the middle panel of Fig. 4. For instance, the 8-column aggregate (dark blue full line), a relatively efficient scatterer, is for a given measured dBZ mapped to a low IWC value, resulting in relatively low IWP overall.
Conversely, the Evans snow aggregate (orange full line) is mapped to high IWC values, resulting in considerably higher IWP.

The retrieved IWP values cover the DARDAR IWP fairly well, though no particle type gives consistent agreement. The large block aggregate gives good agreement at the convective core, while Evans snow aggregate gives better agreement at lower IWP values.

### 4.1.3 Mean IWC distributions

Figure 6 shows the mean retrieved IWC for the different particle types as a function of altitude. The IWC means are derived by averaging the IWC fields, including data from all simulated scenes. The left panel uses MH97, middle F07T, and right D14. The DARDAR IWP is included in the black line in all panels.

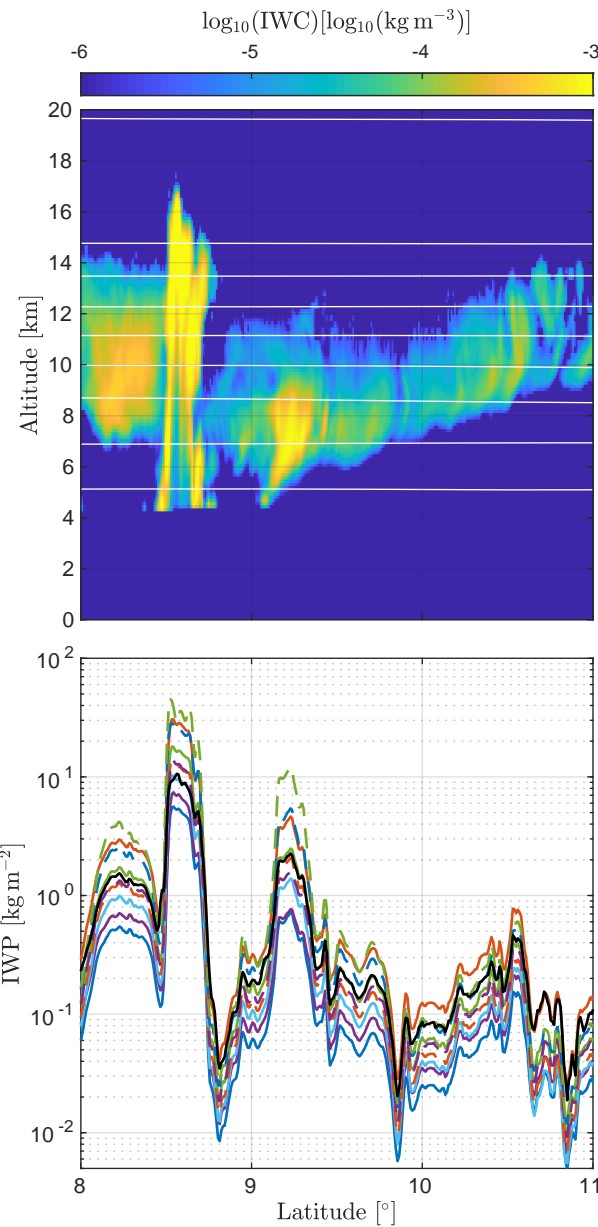

**Figure 5.** Example scene with a deep convective core and some medium to high altitude ice clouds. (Top) DARDAR IWC field and temperature contour lines, going from 270 K to 200 K, in steps of 10 K. (Bottom) Ice water path. The black line represents the DARDAR product IWP, while the coloured lines are from radar inversions assuming different particle types, as indicated by the line color. See legends in Fig. 6 for full description of the lines. F07T is the assumed PSD for all the coloured lines in this figure.

There is a clear difference between the PSDs. MH97 is found to have lower sensitivity to the assumed particle type. At an altitude of 5 km, where the highest amount of ice is found and the uncertainty is the largest, spread is roughly a factor

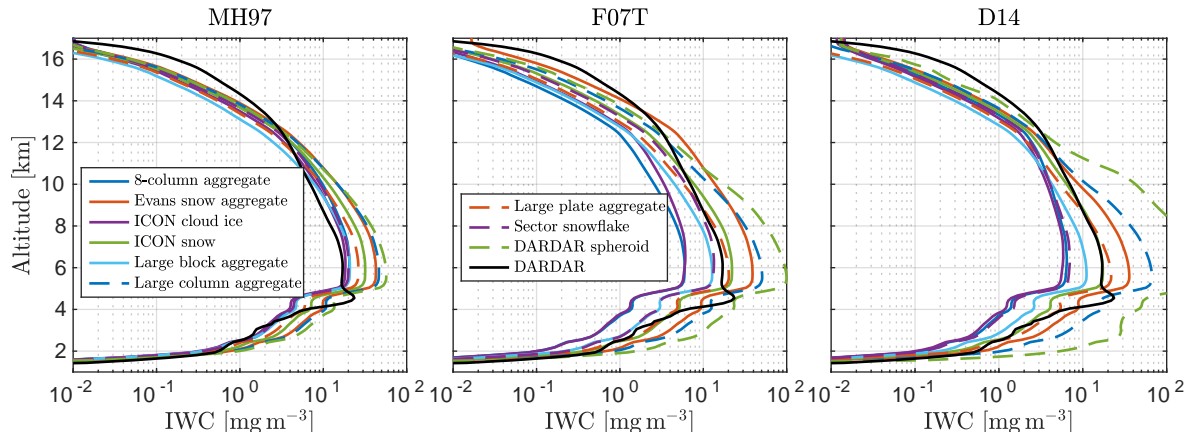

**Figure 6.** Mean IWC as a function of altitude. Assumed PSD is MH97 (left), F07T (middle), and D14 (right).

3 for MH97 and over a magnitude for F07T. However, the lines of the particles are ordered in a roughly the same way for both PSDs. The D14 PSD show similar tendencies to F07T. The exception is the DARDAR-spheroid, which yields noticeably higher values of IWC between 5 to 10 km compared to F07T.

In comparison to the DARDAR IWC, all PSDs seem to underestimate IWC at altitudes above 12 km. The CALIPSO lidar is not included in the inversions and the CPR is not sensitive to thin ice clouds, hence, the inversions put too low IWC at high altitudes. This is also demonstrated in Fig. 5, where retrieved IWP is consistently lower in comparison to DARDAR for clouds with less than $1 \, \mathrm{kg \, m^{-2}}$. Between 5 and 10 km, MH97 tends to higher IWC compared to DARDAR. F07T and D14, on the other hand, cover the DARDAR line fairly well at these altitudes. For both F07T and D14, the large plate aggregate and ICON snow lie close to the DARDAR IWC.

## 4.2 Simulated GMI observations

This section overviews the simulated measurements of TB. First, bulk extinction coefficients are overviewed, useful for the interpretation of simulated TBs. TBs are then shown for the same example scene as presented in Sect. 4.1.2. Finally, simulated TBs are analysed statistically including the results from all simulated scenes.

### 4.2.1 Bulk extinction

Bulk extinction is shown in Fig. 7 as a function of either IWC (left panel) or radar reflectivity (middle and right panel). Extinction is shown in terms of extinction coefficient ratios, i.e., $\gamma_{\mathrm{ext}}$ of given particle divided by $\gamma_{\mathrm{ext}}$ of a reference particle, in this case the 8-column aggregate. The extinction coefficient is given by

$$\gamma_{\mathrm{ext}} = \int_{0}^{\infty} \sigma_{\mathrm{e}}(D_{\mathrm{veq}}) N(D_{\mathrm{veq}}) \, \mathrm{d}D_{\mathrm{veq}}, \tag{5}$$

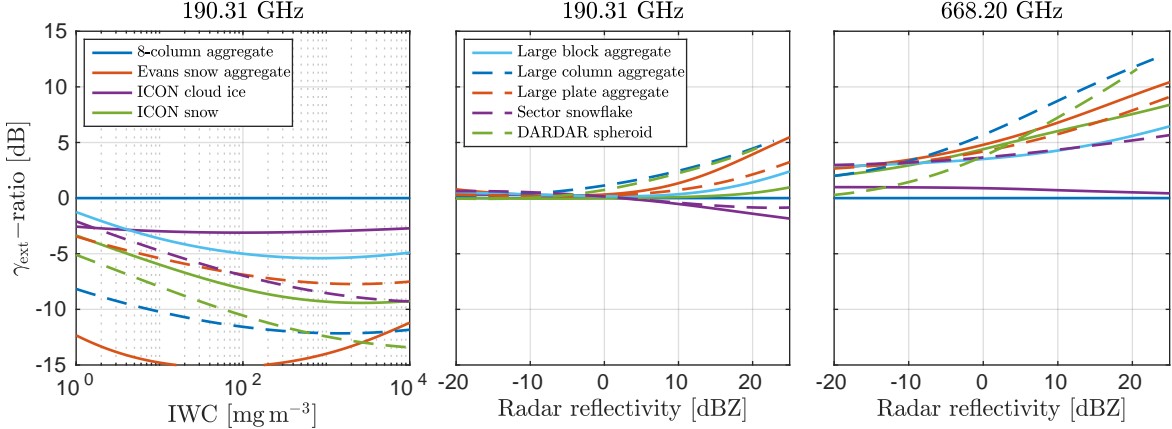

**Figure 7.** Ratios of extinction coefficient $\gamma_{\mathrm{ext}}$ as a function of IWC (left) and radar reflectivity (middle and right). The ratios are calculated by dividing $\gamma_{\mathrm{ext}}$ of given particle model, with $\gamma_{\mathrm{ext}}$ of the 8-column aggregate. Hence, the 8-column aggregate appears as a straight line. Frequency is 190.31 GHz (left and middle) and 668.2 GHz (right). PSD is F07T and temperature is 255 K.

where $\sigma_{\mathrm{e}}$ is the extinction cross-section. Frequency is 190.31 GHz in the left and middle panels and 668.2 GHz in the right. PSD is F07T in all panels, and assumed temperature is 255 K. The spread between particle models in the left panel is large,
over 10 dB, implying that simulated TBs are indeed highly dependant upon assumed particle model (in the IWC retrieval). On the other hand, there is a clear reduction in spread in the middle panel, where extinction is expressed in terms of radar reflectivity. This will partly explain the results shown in the next sections. In the right panel, frequency is 668.2 GHz instead. The spread is slightly larger, with the 8-column aggregate, ICON cloud ice, and DARDAR spheroid particles standing out. Extinction coefficient $\gamma_{\mathrm{ext}}$ as a function of IWC at 668.2 GHz (not shown here), behaves fairly similar compared to the lines
in the left panel.

### 4.2.2 Example scene

Simulated TBs are here presented for the example scene shown in Fig. 5. In Fig. 8, simulations at 190.31 GHz for different combinations of PSD, particle type, and modelling mode (IWC-based versus dBZ-based), are presented. See the figure caption for details on how the data are displayed.
The top left panel assumes F07T as a PSD and makes use of the IWC-based mode. The uncertainty in TB due to particle type assumption is the highest for this combination. For instance, the spread of lines at the peak at 9.2° is almost 60 K. The uncertainty is significantly reduced in the bottom left panel, where the dBZ-based mode is used instead. The effect is similar in the top right panel, where the PSD has been switched to MH97 (but still using the IWC-based mode). In both cases the uncertainty is reduced by a factor of roughly 3. There is also a noticeable shift towards lower TB-depressions when MH97 is
used, due to the higher emphasis towards smaller particles of this PSD. Lowest spread overall is attained in the bottom right

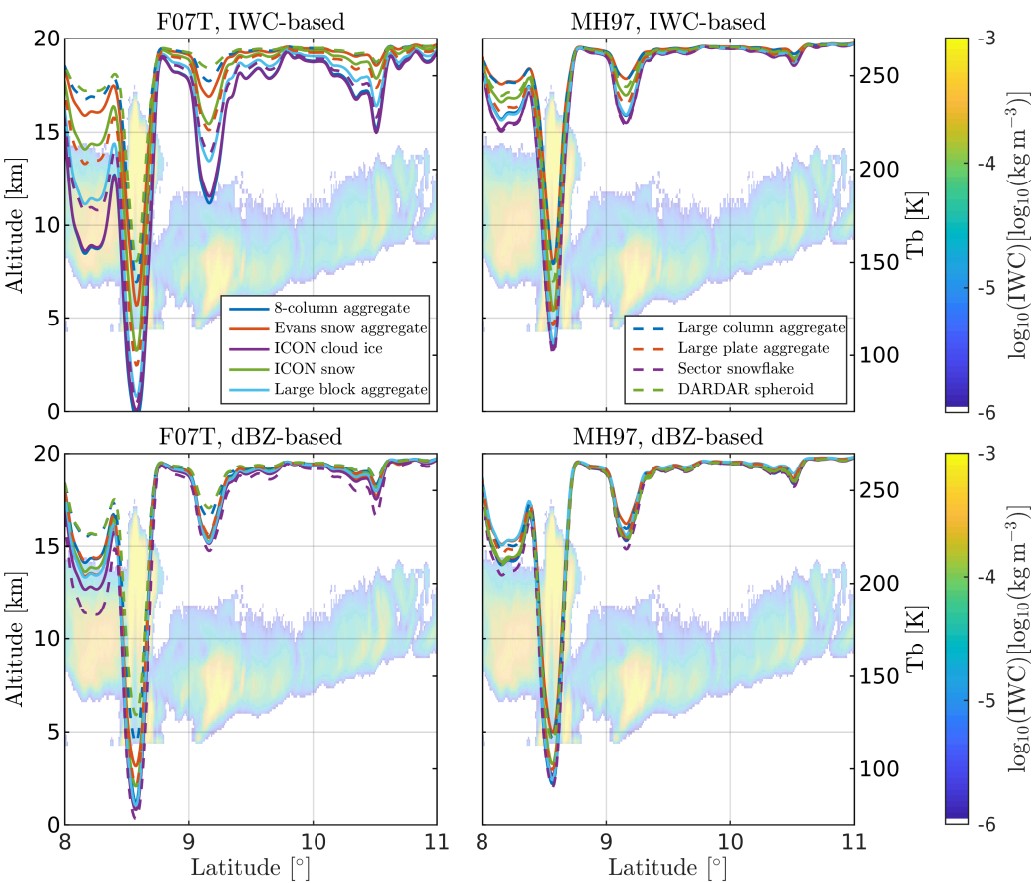

**Figure 8.** Simulated TBs for the same scene as in Fig. 5. The simulations in each panel assumes different PSD and model system mode. Frequency is 190.31 GHz in all panels. Employed modes are: IWC-based (top row) and dBZ-based (bottom row). PSDs are: MH97 (left column) and F07T (right column).

panel, where PSD is MH97 and the dBZ-based mode is employed. Spread is reduced by roughly a factor 6 in comparison to the top left panel. The tendencies discussed here largely holds for the other frequencies as well, but is not shown here.

Figure 9 is similar to Fig. 8, but the used model system and PSD is kept constant for all panels. Instead, different frequencies is displayed in each panel. PSD is F07T and the dBZ-based mode is used, hence the panels can be compared with the bottom

left panel of Fig. 8 where the frequency is 190.31 GHz. The top left panel illustrates TBs at 186.31 GHz and displays similar tendencies to 190.31 GHz, but with lower TB-depression overall. This is a consequence of 186.31 GHz being closer to the centre of a water vapour absorption line, i.e., the TB-depressions are masked by the stronger water vapour absorption at higher altitudes. The simulations at 328.65 and 334.65 GHz shown in top right and bottom left panel, respectively, show similar tendencies. At 328.65 GHz, the TB-depressions are lower compared to 334.65 GHz, especially at the peak at 9.2°. As for

190.31 GHz, this is a consequence of proximity to a water vapour absorption line. Relative to the channels close to 180 GHz,

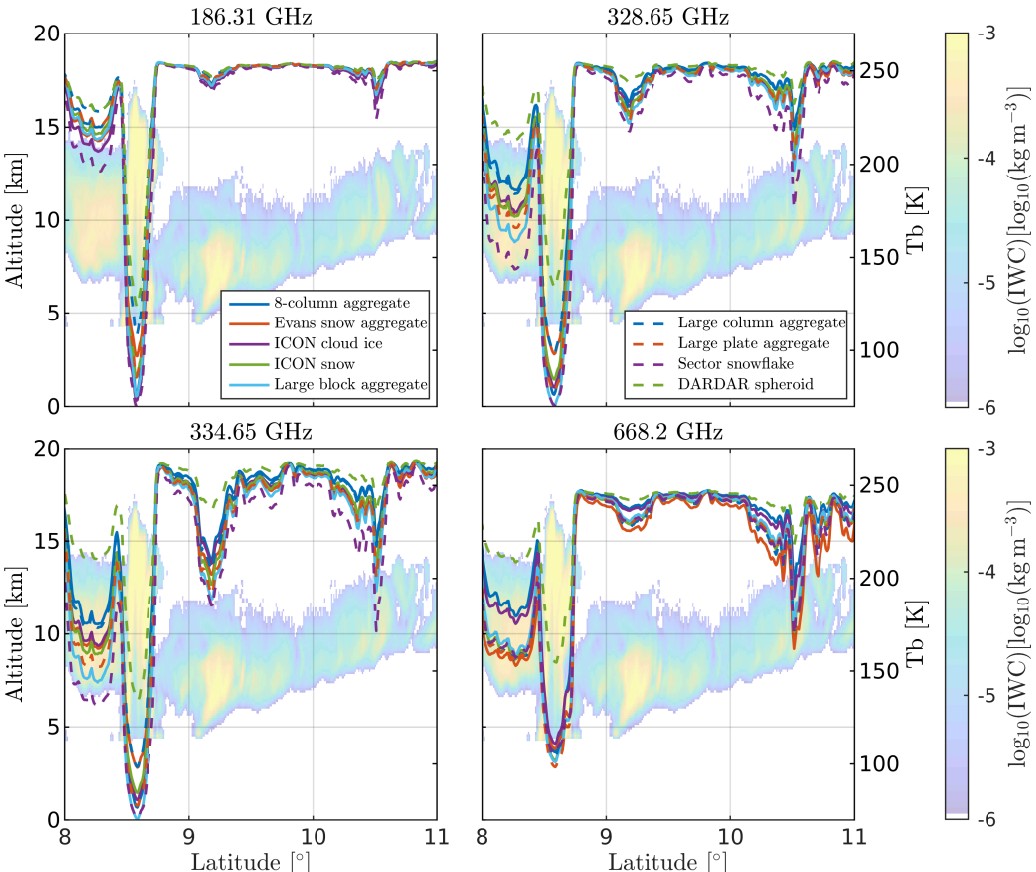

**Figure 9.** As in Fig. 8, but all panels use PSD F07T and the dBZ-based mode. Instead, frequency is different in each panel. Frequencies are: 186.31 GHz (top left), 328.65 GHz (top right), 334.65 GHz (bottom left), and 668.2 GHz (bottom right).

the 300 GHz channels show higher TB-depression for thinner clouds, as evident by the peaks at 9.2 and 10.5°. However, the uncertainty is fairly similar in magnitude, if one excludes the DARDAR-spheroid particle in the dashed green line. The DARDAR-spheroid is a clear outlier at the higher sub-millimetre frequencies.

In the bottom right panel, simulations are shown for 668.2 GHz. This channel shows the highest TB-depressions among all the channels for high altitude thin clouds, as evident by the peaks at 10.5° and higher. There also seem to be a more distinct separation between different particle types. As for the 300 GHz channels, the DARDAR-spheroid results in relatively low TB-depressions. Furthermore, the 8-column aggregate and the ICON cloud ice are also distinguished, having clustered into a group of intermediate values of TB. The remaining particle types have clustered into a large group with high TB-depressions.

### 4.2.3 TB-distributions

This section gives a statistical overview of simulated TBs. Figure 10 displays TB occurrence frequencies for different combinations of particle types, PSDs and model systems. Simulations using the D14 PSD are omitted here, since they produce fairly similar results compared to MH97. Black lines represent GMI data (see Sect. 2.2) at 190.31 GHz, for comparison. The peak at around 270 K corresponds to simulations at clear sky cases, which is the most frequent scenario. Lower TB-values correspond to increasingly high-altitude and thick clouds. The lowest TB-values are typically associated to deep convective cores, as seen

in Fig. 8 for example. This was verified by partitioning the cold simulated observation by the CloudSat classification product, revealing that around 70 % of TBs below 200 K are categorized as deep convection.

The top left panel, which uses F07T and the IWC-based mode, shows the largest spread in lines. On the other hand, the lines cover the GMI measurements in the black line fairly well. In the top right panel, PSD is switched to MH97, showing a clear reduction in uncertainty (as in Fig. 8). However, for MH97 there is a tendency towards underestimating TB in comparison to

440 GMI.

In the bottom panels, the dBZ-based mode is used instead, showing a clear decrease in uncertainty for both PSDs. Both PSDs agree well to GMI above 170 K, showing no significant bias. It is difficult to distinguish which particle models agree best to GMI by eye; the exception being the DARDAR-spheroid in the left bottom panel. However, the forward simulations tend to produce a relatively large amount of very low TBs. Only the large column aggregate agrees with GMI at these cold

TBs, for both F07T, MH97, and D14. As already mentioned, the simulated distributions using the D14 PSD (not shown here) are similar to MH97 when the dBZ-based mode is used.

By visual inspection, agreement to GMI (channels explored) is good, indicating the robustness of the forward simulations at these frequencies. However, there is a mismatch close at the peak of the distributions, indicating that clear-sky observations are not matching perfectly. The fact that, for most particle and PSD combinations, distributions predict significantly colder

temperatures than GMI, indicates that simulated observations of deep convection are not matching GMI perfectly. Simulated distributions at 166 and 186.31 GHz (not shown here) behave very similar in comparison to the simulations at 190.31 GHz.

At higher simulated frequencies, there are currently no measurements available for comparison. At 328.65 and 334.65 GHz (not shown here) spread of the lines is slightly higher in comparison to 190.31 GHz, for all PSD and model mode combinations. The exception is the DARDAR-spheroid which for some combinations produce significantly lower amount of cold TBs.

At 668.2 GHz, the picture is somewhat different. Figure 11 overviews simulated TBs at 668.2 GHz. The layout is similar to Fig. 10, but simulations using the IWC-based mode are omitted, and MH97 is switched out for D14 in the right panel. In contrast to the dBZ-based simulations at 190.31 GHz, a clear spread between particle models is observed at intermediate TB-values. In essence, the simulations at 668.2 GHz demonstrates higher sensitivity to particle type assumption than for 190.31 GHz, with some exceptions. The DARDAR-spheroid is the most significant outlier, generally resulting in very warm distributions

compared to the other particles.

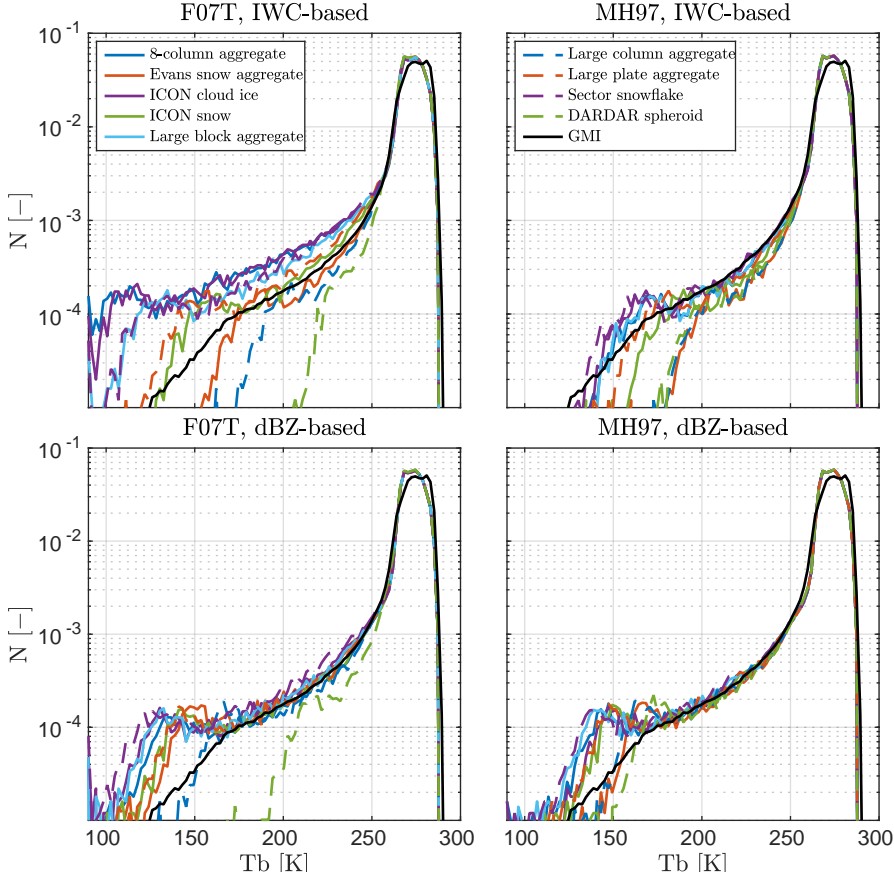

**Figure 10.** Occurrence frequencies of TBs at 190.31 GHz assuming various particle models. Curves are normalized to have unit area. The different coloured lines represent the particle type used in the forward simulation. The black line is derived from GMI measurements. Used modes are: IWC-based (top row) and dBZ-based (bottom row). PSDs are: MH97 (left column) and F07T (right column).

### 4.2.4 Overview of performance

In order to provide an overview of the performance of the particle models and PSDs, differences between simulated and GMI-observed mean TBs are calculated. Because collocated passive and active observations were not used in this study, it is not possible to perform a quantitative error analysis. Hence, we are limited to comparing statistics of the simulations and GMI 465 observations.

The mean TBs are derived as

$$\overline{T}_{\mathrm{b}} = \frac{\int_0^{240} T_{\mathrm{b}} N(T_{\mathrm{b}}) \, \mathrm{d}T_{\mathrm{b}}}{\int_0^{240} N(T_{\mathrm{b}}) \, \mathrm{d}T_{\mathrm{b}}}, \tag{6}$$

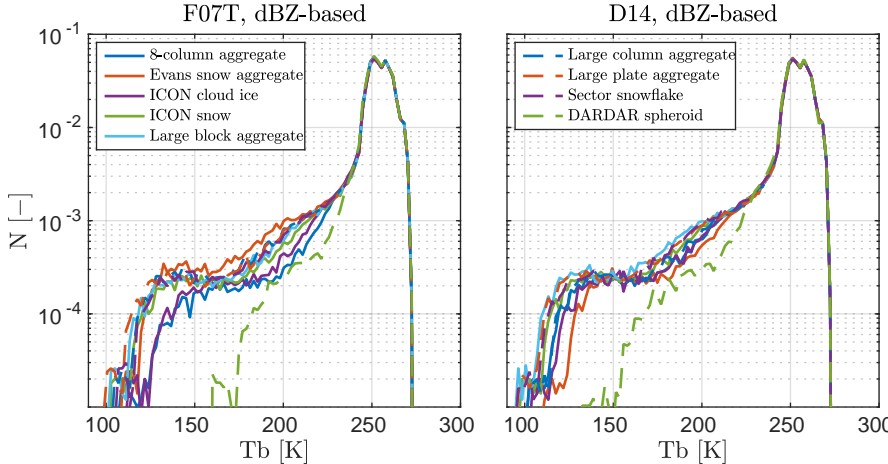

**Figure 11.** As the bottom panels of Fig. 10, but for 668.2 GHz and the D14 PSD is used to the right.

where $N$ is the distribution of TBs. Clear-sky cases are removed by only considering TBs below a threshold of 240 K. The difference in mean TBs is then

$$\Delta\overline{T}_{\mathrm{b}} = \overline{T}_{\mathrm{b}} - \overline{T}_{\mathrm{b,GMI}}. \tag{7}$$

Table 2 shows $\Delta\overline{T}_{\mathrm{b}}$ for each GMI frequency, particle model and PSD combination. The particle and PSD combinations generally result in negative TB differences, larger than 10 K for at least one channel. One exception is the DARDAR spheroid, that tend to positive differences and surprisingly results in fairly low TB differences when combined with MH97. The large column aggregate typically result in the lowest TB differences for all PSD and frequency combinations. The Evans snow aggregate combined with MH97 yields the lowest TB differences by a small margin, but it does not perform well with the other PSDs. Focusing on the PSDs, F07T yields the largest TB differences for most frequencies and particle models, while MH97 typically yield the smallest. As previously explained, F07T puts high emphasis on larger particles, explaining the stronger TB-depressions compared to MH97. In summary, the large column aggregate and the Evans snow aggregate, combined with MH97 result in the lowest TB differences.

## 5 Discussion

First of all, comments on the robustness of the simulations are in order. As shown in Fig. 6, the radar inversions agree fairly well with the DARDAR ice water content (IWC) product. However, it was also noted in Sect.4.1.3 that the radar inversions put relatively low IWC at high altitudes compared to DARDAR, as a consequence of the exclusion of lidar measurements in the inversions and the insensitivity of CloudSat to thin ice clouds. Calculated IWC-dBZ-relationships also agree well with the relationship of Protat et al. (2016) (see Fig. 4), which are based on collocated airborne radar and in-situ measurements. MH97 tend to overestimate IWC for most particle models in comparison to Protat. However, as mentioned in Sect. 4.1.1, the

**Table 2.** Differences between the means of simulated and GMI-observed TBs (see Eq. 7), in units of Kelvins. Only TBs below 240 K are considered, essentially filtering out clear-sky cases.

| Particle model | D14 | | | F07T | | | MH97 | | |
| --- | --- | --- | --- | --- | --- | --- | --- | --- | --- |
| | 166 GHz | 186 GHz | 190 GHz | 166 GHz | 186 GHz | 190 GHz | 166 GHz | 186 GHz | 190 GHz |
| 8-column aggregate | -19.6 | -8.84 | -14.0 | -28.6 | -13.6 | -22.9 | -15.1 | -8.55 | -17.5 |
| Evans snow aggregate | -14.9 | -3.85 | -10.3 | -14.3 | -13.4 | -14.1 | 0.54 | -3.54 | -3.74 |
| ICON cloud ice | -15.6 | -9.80 | -14.3 | -25.6 | -16.9 | -20.6 | -12.4 | -8.59 | -14.7 |
| ICON snow | -15.5 | -6.94 | -13.2 | -19.4 | -10.7 | -19.2 | -10.0 | -8.53 | -13.2 |
| Large block aggregate | -21.4 | -10.4 | -17.8 | -25.0 | -16.4 | -21.7 | -15.7 | -11.7 | -18.1 |
| Large column aggregate | -7.82 | -3.71 | -7.14 | -1.57 | -4.53 | -4.71 | -0.09 | -4.72 | -3.82 |
| Large plate aggregate | -22.8 | -11.1 | -16.4 | -18.8 | -10.5 | -18.2 | -11.5 | -7.29 | -14.9 |
| Sector snowflake | -12.7 | -9.56 | -16.8 | -19.2 | -17.4 | -21.4 | -17.6 | -13.1 | -19.6 |
| DARDAR spheroid | 15.7 | 8.91 | 8.56 | 29.6 | 14.6 | 17.4 | 8.12 | -0.81 | 0.50 |

Protat relationship is likely influenced by azimuthally oriented particles, which enhances zenith radar backscattering (Hogan et al., 2012). Furthermore, Fig. 10 indicates that the agreement of the forward simulations to the GPM Microwave Imager (GMI) observations on a statistical level is good. Main differences appear at around 270 K, i.e., clear sky cases. It is found that the difference in mean TB between simulated and GMI clear sky observations is about -0.1, -4.7 and -2.6 for 166, 186.31 and 190.31 GHz, respectively. The negative biases could be an indication that ERA-Interim puts too high relative humidity at high altitudes in the considered region, likely related to the prevalence of convection. The higher bias at 186.31 compared to 190.31 GHz further supports this claim, since water vapour absorption is stronger at 186.31 GHz.

Furthermore, no simulated distribution of simulated brightness temperature (TB) manages to fully reproduce the GMI TB distribution below 170 K. Most simulated distributions show a relatively sharp drop in frequency at relatively cold TBs in comparison to GMI. For many cases, the drop is preceded by a slight increase in occurrence frequency. The problem is most likely connected to deep convection. One possible explanation is that GMI measures warmer TBs because of azimuthally oriented particles. Since totally random particle alignment implies a decrease in zenith radar back-scattering in comparison to azimuthal alignment, an increase in retrieved IWC is expected. The increase in IWC should then lead to an enhancement in the simulated TB-depressions, given that forward simulations (of intensity) are not strongly affected by particle orientation at an incidence of 53°, as indicated by the results of Brath et al. (2019). Testing this hypothesis requires azimuthally oriented data, which at the time of this study was not publicly available. On the other hand, previous studies also indicate that ice particles in deep convection are likely totally randomly oriented dense particles such as hail and graupel, inferred from the low observed polarization signals (Gong and Wu, 2017; Brath et al., 2019). Further investigation is needed in order to explain these discrepancies.

Regarding sensitivity to assumed microphysics of the forward simulations: Simulated TB-distributions are indeed sensitive to the assumed particle model at all frequencies, as evident in the top panels of Fig. 10, where the IWC-based approach was

employed. In short, in the IWC-based approach, fixed IWC fields (i.e., from the DARDAR product) are used as input to the forward simulations. Similarly, the radar retrieved IWC fields (see Fig. 6) show significant sensitivity to particle model as well, resulting in spread of over one order of magnitude, depending on the assumed PSD. Whether the application is assimilation or retrievals, the results show the importance of correct representation of the ice microphysics in radiative transfer (RT) modelling, for both passive measurements at frequencies above $180\,\mathrm{GHz}$ and radar measurements at $94.1\,\mathrm{GHz}$. As indicated by Fig.1, it is predominantly larger particles (roughly $1\,\mathrm{mm}$ and larger), that contribute to the observed discrepancies for both active and passive measurements.

However, analysis of simulated TB-distributions reveal little on the performance of tested particle models. The other model mode, the dBZ-based approach, uses CloudSat dBZ fields as input directly (see Fig. 3 for a schematic overview), ensuring that microphysical assumptions are consistent both for the active and passive sensor. When using the dBZ-based mode, simulated TB-distributions reveal little sensitivity to the assumed particle model, as seen in the bottom panels of Fig.10. This is mainly explained by a compensation effect, illustrated in Fig. 7. The extinction coefficient $\gamma_{\mathrm{ext}}$, arguably the most important parameter in microwave radiometry, is indeed sensitive to hydrometeor particle type, as shown in the left panel. When $\gamma_{\mathrm{ext}}$ is expressed as a function of radar reflectivity (see the middle panel), a more compact collection of lines is attained. The extinction coefficient at $190.31\,\mathrm{GHz}$ is to some extent less dependent upon particle type when expressed in radar reflectivity, in comparison to IWC. For instance, the 8-column aggregate tends to result in very low IWC (Fig. 4), since it is a very efficient back-scatterer. However, since it is an equally efficient attenuator (Fig. 7, left panel), the resulting TBs end up at similar values compared to the other particles.

At intermediate TBs (between 170 and $250\,\mathrm{K}$), most particles produce TB-distributions at 166, 186.31 and $190.31\,\mathrm{GHz}$ that lie close to GMI. The only consistent noticeable outlier is the DARDAR-spheroid, which tend to very warm TB-distributions when combined with F07T and D14 (not shown). As discussed above, disagreement to GMI is mainly visible below $170\,\mathrm{K}$, with most particles resulting in very cold TB-distributions. No PSD and particle combination is able to fully mimic the GMI curve. Nonetheless, for some cases the distributions end up fairly close. For instance, in the case of MH97 that results in relatively low spread between distributions, the large column aggregate, Evans snow aggregate, and the DARDAR spheroid lie fairly close to GMI. This is also reflected in Tab. 2, showing relatively low differences in mean TBs compared to GMI for these combinations. But only the large column aggregate is found to lie close to GMI for all PSD and frequencies (bottom panel of Fig. 10 and Tab. 2). Interestingly, the large column aggregate is among the weakest scatterers (see Fig. 1), which is also the case for the DARDAR spheroid. The difference is that the DARDAR spheroid is also an excessively weak back-scatterer at large particle sizes. At high IWC, the large column aggregate tend to lower dBZ-values in comparison to Protat et al. (2016), while at intermediate IWC ($\approx 10^2\,\mathrm{mg\,m^{-3}}$) a good agreement is found when employing F07T and D14 (see Fig. 4). As mentioned previously, assuming totally random orientation should yield lower dBZ in comparison to Protat, i.e., the disagreement is to some extent expected. Regarding the radar inversions, in comparison to the DARDAR IWC product, they put fairly high IWC at mid-altitudes (see Fig. 5 and 6). However, since the DARDAR product does not account for cloud attenuation of the radar signal, DARDAR IWC is expected too be underestimated to some degree.

All PSDs managed to largely reproduce the GMI distribution, except for the discrepancies already discussed. D14 tend to produce marginally warmer distributions compared to GMI at 166, 186.31 and 190.31 GHz (not shown). The same can be said for MH97, mostly for 186.31 GHz (not shown), but also slightly for 190.31 GHz (compare to F07T at 260 K in bottom panels of Fig. 10). F07T shows no obvious bias compared too GMI, but is instead the PSD for which the spread between particle models is the highest. Overall, MH97 could be argued to perform the best, in terms of agreement to GMI, typically resulting in the smallest differences in mean TB in Tab. 2. It also agrees fairly well with the dBZ-IWC relationship of Protat et al. (2016) (see Fig. 4), especially if particle orientation is taken into consideration.

Overall, the attempt to constrain assumed microphysics using CloudSat and GMI measurements produced fairly inconclusive results. Best agreement to GMI was found for the large column aggregate and Evans snow aggregate when combined with the MH97 PSD (see Tab. 2). However, the failure of the forward model to mimic the GMI distribution at cold TBs suggests this conclusion should be taken with a grain of salt. On the other hand, the DARDAR spheroid is revealed as a fairly poor representation of ice considering its tendency towards very warm TB-distributions, especially at the higher sub-millimetre frequencies. No judgement with respect to its performance in radar and lidar retrievals is made here; however, for passive radiometry at frequencies above 180 GHz, it deviates significantly from the more realistic models. Also, compared to the other particles and the DARDAR IWC product, it produces significantly higher IWC in the radar inversions, especially using the D14 PSD (see Fig. 6). As a test, forward simulations assuming the DARDAR-spheroid, D14, and DARDAR-retrieved IWC-and $N_0^*$ fields were performed, essentially to check whether the DARDAR product and its assumptions could reproduce GMI measurements. A small improvement is gained compared to using the a priori determined $N_0^*$ fields (i.e., using Eq. 3). However, the resulting TB-distribution was still found to be significantly warmer compared to GMI. Whether this disagreement is due to the particle model or the DARDAR retrievals themselves, is difficult to conclude based on this paper alone. However, it does highlight the DARDAR spheroids inability to yield consistent results in a combined active and passive microwave setting. A similar conclusion was reached by Kulie et al. (2010) for soft spheres. As final disclaimer, the scattering data of the DARDAR spheroid was (as stated in Sect. 2.6) reproduced according to info given in Delanoë et al. (2014) and using the T-matrix method, to the best of our abilities. In order to test our implementation of the DARDAR spheroid, radar reflectivity fields were calculated for the scene in Fig. 5 assuming the DARDAR spheroid, D14, and DARDAR retrieved IWC- and $N_0^*$ fields, (without accounting for radar attenuation, as is the case for the DARDAR retrievals). The reflectivity fields were then compared to the corresponding CloudSat measurements. Agreement is generally acceptable; difference is generally less than 2 dBZ. However, errors over 5 dBZ were found in the convective core. Hence, we acknowledge that some inconsistencies in our implementation of the DARDAR spheroid model is a possibility.

Better agreement to observations could of course also be achieved with a more advanced microphysical scheme. For instance, the CloudSat cloud classification product (2B-CLDCLASS) can be used to guide microphysical assumptions in specific cloud scenarios. This was omitted in this study due to the difficulty in performing a quantitative error analysis of the simulations. Instead, this study evaluates the ability of each particle model and PSD combination to work as an "one-size-fit-all" microphysical parametrization.

Finally, as a comment on the significance of this study for the upcoming Ice Cloud Imager (ICI) mission: Simulated distributions at frequencies above 300 GHz revealed a significantly higher degree of sensitivity to particle model for the dBZ-based simulations, than what was found at the lower frequencies. As seen in the right panel of Fig. 7, the relationship between $\gamma_{\mathrm{ext}}$ and radar reflectivity at 664 GHz is indeed more sensitive to the particle model than for 190.31 GHz. Hence, there is indication that the methodology presented in this paper, with input from ICI and a high frequency radar (e.g, CloudSat CPR or the upcoming EarthCARE CPR), could a be useful tool for refining particle models. To some extent, this study also supports the notion of combined sub-millimetre radiometer and high frequency radar measurements from a retrieval point of view, which, for example, could potentially be used to retrieve PSD moments (Pfreundschuh et al., 2019).

## 6   Conclusions

This paper has presented simulated brightness temperature (TB) measurements in the tropic Pacific ocean by the GPM Microwave Imager (GMI) and the upcoming Ice Cloud Imager (ICI), based on synthetic scenes using CloudSat measurements as a basis. Forward passive simulations use either ice water content (IWC) fields from the DARDAR product (the IWC-based approach) or first convert CloudSat dBZ-values to IWC fields assuming microphysics (i.e., particle model and size distribution) that are fully consistent with the forward simulations (the dBZ-based approach). The main goal was to evaluate and constrain a selection of particle models from the ARTS (Atmospheric Radiative Transfer Simulator) single scattering database (Eriksson et al., 2018), by analysing their ability to model observations by both CloudSat and GMI. The advantage of the method is that it does not require collocated passive and active measurements, which are scarce around the equator. The limitation is that it does not allow for a quantitative error analysis. Instead, one is limited to approximate comparisons of statistics and visual interpretation. Main conclusions are as follows:

– Simulated TB distributions at 166, 186.31 and 190.31 GHz overall agree well with GMI. The biggest issues occur under clear-sky conditions, like due to biases in ERA-interim reported humidity fields, and for the most severe cases of deep convection, possibly due to the fact that azimuthal particle orientation is not considered in the radiative transfer model employed here.

– The simulated distributions at GMI frequencies proved to be fairly insensitive to the assumed particle model and particle size distribution (PSD), at least at intermediate TBs. Most particle models seems to yield scattering consistent to both CloudSat and GMI observations. Hence, it was not possible to find a particle model that clearly performs better than the others. The large column aggregate and Evans snow aggregate combined with the MH97 PSD seem to be best overall, since they reproduce the very cold TBs of GMI fairly well. The other particles and PSD combinations typically result in too cold distributions, as mentioned previously. The exception is the soft-spheroid particle model (used in the DARDAR product), generally producing too warm TBs. Interestingly, it agrees well with GMI when combined with the MH97 PSD.

- In comparison to the dBZ-based approach, the IWC-based simulations were found to be highly sensitive to the assumed particle model and PSD, including at the higher sub-millimetre frequencies (328.65, 334.65, and 668.2 GHz). It is mainly the larger particles that contribute to the discrepancies in simulated TBs. This highlights the importance of correct microphysical representation in radiate transfer modelling of high microwave and sub-millimetre passive measurements.

- Assumed particle model also has a significant impact upon IWC fields retrieved by radar. Differences of over one order of magnitude are found in retrieved IWC-fields between employed particle models. Also here does the soft-spheroid model stand out, resulting in significantly higher IWC compared to the other particles, being a relatively weak scatterer.

- Of tested PSDs, the one by McFarquhar and Heymsfield (1997) results in TB distributions that for most particle models, most closely resemble the distribution of GMI measurements. It also produces IWC-radar reflectivity relationships that agree fairly well with the empirical relationship by Protat et al. (2016).

While this study proved inconclusive with respect to its main goal, it still serves as a demonstration to the methodology used for evaluating microphysics and highlights the importance of continued work. The simulations at sub-millimetre frequencies are promising for the sake of the upcoming ICI mission, the first planned operational satellite aimed towards measuring atmospheric ice at such frequencies. It will offer additional sensitivity to smaller ice particles, as indicated by the simulations performed in this study. This study shows that, using the presented methodology, combined measurements by ICI and a 94 GHz cloud-profiling radar could potentially be used to constrain microphysical parametrizations, i.e., particle shape, PSD, and possibly orientation.

The methodology should also be extended to consider passive infrared measurements (e.g., IIR aboard CALIPSO or MODIS on Aqua), providing additional microphysical constraints (to the PSD in particular) and sensitivity to smaller ice crystals that radar is lacking. Information is typically only gained at the cloud top environment at IR frequencies, however, improved constraining of microphysics at these level would benefit analysis of concurrent levels as well. This was omitted in this study, since the infrared spectrum is not covered by the ARTS scattering database. At these high frequencies, the discrete dipole approximation (DDA), traditionally used at microwave wavelengths, is not applicable for calculating the scattering properties, and surface roughness and inhomogeneities such as air bubbles play an important role (Tang et al., 2017). So far, only the database by Yang et al. (2005) covers such frequencies (was extended by Ding et al. (2017) to also include microwaves). Future studies should also consider scattering data describing melting hydrometeors (Johnson et al., 2016; Ori et al., 2014) and oriented ice particles (Brath et al., 2018).

*Code availability.* Available upon request.

*Author contributions.* RE implemented the simulations and performed the data analysis, with advice and assistance from PE. RE also wrote the manuscript, with feedback from PE and SP. Finally, PE and SP contributed with code needed to implement the simulations.

*Competing interests.* The authors declare that they have no conflict of interest.

*Acknowledgements.* First of all, thanks to the reviewers for their fair and insightful feedback. The authors would also like to thank their colleagues at the Department of Space, Earth and Environment, Chalmers University of Technology, who provided suggestions and support
to the data analysis. Special thanks goes to Vasileios Barlakas who also provided detailed feedback on the manuscript. Finally, this study was financially supported by the Swedish National Space Agency (SNSA) under grant 150/14.

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
