# Peer review of "Using passive and active observations at microwave and sub-millimetre wavelengths to constrain ice particle models"

_Atmospheric Measurement Techniques, 2019_

## Referee Comment (RC1) · Anonymous Referee #1 · 10 Sep 2019

This is a very thorough attempt to validate and constrain the large number of particle models available in the recently published Eriksson et al. (2018) single scattering DDA database. This database, known as the ARTS scattering database, contains 34 particle models over 34 frequencies (1 - 886.4 GHz). This study addresses the complex question which models are the best representations of ice hydrometeors? In order to do so, a combined active and passive radiative transfer model framework is employed under a combination of particle models and particle size distributions. CloudSat observations are converted to simulated brightness temperatures at GMI (only 186.31 and 190.31 GHz) and ICI (328.65, 334.65, and 668.2 GHz) channels. Simulated TBs are compared with the real GMI observations, but the study concludes it is difficult under GMI frequencies to choose a particular particle model over another one, although ICI

simulations show the potential these observations will have in the near future to tackle the central question of this study. The publication is fit for publication with only a few points to be discussed or considered.

General comments

* One of my main comments is why the GMI 89 and 166 GHz channels were not simulated. Trying to address the representativeness of ice particle models, excessive scattering at the lower microwave frequencies should be avoided and these channels could help. These channels would also give a very complete frequency-wise analysis of the most up to date database coupled with ARTS, which is extensively used. Is this outside the scope of this paper?

* Similarly, with the 243 GHz channel in ICI.

Specific Comments (individual scientific questions/issues)

Abstract: (L13) Could you please elaborate on what is meant by "a compensation effect between bulk extinction at passive frequencies and radar reflectivity." L168: For completeness perhaps it is worth including a comment about the suitability / experience when using the other scattering solvers in ARTS in the sub-millimetre range. Are there other publications where ARTS is used in that range? L193: What is the definition of aspect ratio? L194: What sort of mixture was used? Ice in air, or air in ice? L208: "Above 500 $\mu$m, most of the particles, except the DARDAR spheroid and the 8-column aggregate, have aggregated to a single cluster of lines." Differences are still important. L246: "Overall, while smaller particles are more numerous, intermediately sized particles dominate in terms of scattering impact." From the text only the MH97 PSD has more numerous smaller particles, the other PSDs have higher emphasis on larger particles. So this sentence is a confusing. L267: Also in the importance of consistency when making assumptions throughout the chain of simulations. L285: Please discuss what is meant by "59 CloudSat orbits could be found and selected as references for the synthetic scenes". How coincident in time and space are you requiring CloudSat

to overpass the GMI footprint. L289: By LWC do you mean cloud droplets? Please specify. I am assuming this because a few lines later you say you retrieve Rain Water content too. L291: Please also refer the reader to section 3.2. L380: simulated TBs are indeed highly dependant upon assumed particle model (in the IWC retrieval). L393: lower TB-depressions (i.e., warmer simulated TBs). These are expected for the MH97 PSD since it's inclined to favour smaller particles? L400: Don't you mean the 186 GHz channel is closed to the centre of the water vapour line? L402: 328 vs 334. Why especially different for the 9.2 degree? L429: agreement to GMI (channels explored) is good. L433: you mention that there are a few exceptions. It would be nice to include them in the text. L438: Figure 11: why do you switch from MH97 to D14? L458: I don't understand the comment about azimuthally oriented particles. If there were azimuthally oriented particles, shouldn't the TB depressions be actually larger hence even colder temperatures? You would increase retrieved IWC but you would be simulating colder too? This is the first concluding remark which needs revision, no horizontally aligned DDA shapes have ever been simulated. L490: It would have been nice to evaluate this weak scatterer at the lower GMI frequencies. L520: I didn't catch the discussion made about the DARDAR spheroid. L544: fairly insensitive at 190 GHz. Figure 11 (specially using F07T shows differently) Figure 10. It is interesting that the soft sphere with MH97 and dBZ-based mode, isn't worst than the sector snowflake. L464: simulations performed in this study (specially at 668.2 GHz)

Technical corrections / Minor issues

L165: Suggesting changing the phrase "It is intended to be as general as possible. For instance, radiation is described using the full Stokes vector notation and in terms of usage it behaves as a scripting language" to "It is intended to describe radiation using the full Stokes vector notation in the most general manner possible, allowing a large amount of user input flexibility. In itself ARTS behaves as a scripting language on its own." L184: I should read "A particle mixture consists of pristine crystals" without the "a". L346: Please mention that the colored lines represent IWP from radar inversions

(for the F07 PSD) L390: Instead of "9.2" use of "9.2 latitude" (same in line 402) L392: The effect is similar in the top right panel, where the PSD has been switched to (used is the) MH97 (but still using the IWC-based mode). In both cases the uncertainty is reduced by a factor of roughly 3 (when the dBZ-based mode is used), L437: you use Figure instead of Fig. like in the rest of the paper. L472: reveal little on the performance of (the) tested particle models L495: Please review "they puts fairly [. . .]" formulation/grammar L500: typing error. "too" should be "to".

---

## Referee Comment (RC2) · Anonymous Referee #2 · 3 Oct 2019

The authors provide radiative transfer (RT) simulations at microwave and submilletre frequencies and demonstrate both the sensitivity to microphysical assumptions (ice/snow particle model and the particle size distribution (PSD)) and compare RT simulations to Global Precipitation Measurement Microwave Imager (GMI) observations in the tropics. The authors also present simulations relevant for the upcoming Ice Cloud Imager (ICI) sensor.

The authors tackle a very complicated topic given the considerable variability of cloud microphysical composition and its subsequent effect on upwelling microwave brightness temperatures. They adopt a combined radar (CloudSat) and radiometer (GMI) framework to constrain microphysics and show that this strategy is preferred to methods that do not leverage radar observations. While this strategy has been adopted in

previous studies and is not itself unique, the current study uses state-of-the-art particle model scattering databases and numerous PSD parametrisations to move the scientific needle forward on this topic. Simulations at ICI frequencies are also a novel concept that provide beneficial insight for eventual retrieval development for this instrument. While no particular ice model is shown to offer optimal results when RT simulations are compared to observations, this study provides useful guidance on PSD and ice model combinations that can be used in future retrieval research. The authors also readily acknowledge that no particular ice model can realistically be applied as a universal answer, but illustrate important sensitivity studies that can be used to propel further research on this topic.

The manuscript is well-organized and written in an easy-to-follow manner. Figure quality is excellent. The manuscript can be published largely intact, but I suggest a few minor methodological description improvements and other random suggestions that will hopefully allow the authors to further refine the manuscript. I do not see any obvious or fatal scientific flaws regarding the study design and interpretation of the results. My only comment that could be considered as something more than a minor issue regards other possible options to provide further quantitative analyses that might be interesting to the community. I do not classify the suggestions as mandatory, but hopefully will spur the authors to find further creative ways to tabulate their comparisons to observations. Specific comments are outlined below.

Title: Since 664 GHz is considered in this study, maybe consider adding submillimetre to the title to better advertise the ICI applications? This suggestion is purely semantics, but this study extends beyond the wavelengths typically associated with microwave radiometry. Some remote sensing specialists designate submillimetre wavelengths as a distinct category occupying the space between microwave and infrared, while others may consider ICI-like frequencies as part of the microwave spectrum. Another option is to specifically include sensors like GPM and ICI in the title. This suggestion is not mandatory but is something the authors should consider to better advertise the novel

[Figure]

ICI-related content.

Line 44: Should MLS, SMILES, and Odin-SMR acronyms be explicitly written?

Lines 58-59: A reference or references might be beneficial to prove that soft spheroid models produce good results at single frequencies.

Lines 276-278: A CloudSat-GMI coincident dataset exists, but as the authors mention, tropical coincidences are limited. Would it be worth highlighting how few coincident observations exist compared to higher latitudes? The Rysman et al. (2018) manuscript gives a quantitative analysis of global CloudSat-GMI coincidences for snowing observations (see Figure 2). At the very least, the Rysman et al. (2018) manuscript could be referenced to illustrate this point without the authors calculating their own statistics.

Rysman, J.-F.; Panegrossi, G.; Sanò, P.; Marra, A.C.; Dietrich, S.; Milani, L.; Kulie, M.S. SLALOM: An All-Surface Snow Water Path Retrieval Algorithm for the GPM Microwave Imager. Remote Sens. 2018, 10, 1278.

Line 312: Please provide more information regarding gaseous and cloud liquid water absorption methodology used in the RT simulations. This information will allow other investigators to better replicate the study. Studies have also indicated RT variability using standard water vapour continuum and cloud liquid water absorption models (e.g., Turner et al 2009, Kneifel et al. 2014 and others), so knowing what absorption models were used is essential information.

D. D. Turner, M. P. Cadeddu, U. Lohnert, S. Crewell and A. M. Vogelmann, "Modifications to the Water Vapor Continuum in the Microwave Suggested by Ground-Based 150-GHz Observations," in IEEE Transactions on Geoscience and Remote Sensing, vol. 47, no. 10, pp. 3326-3337, Oct. 2009. doi: 10.1109/TGRS.2009.2022262

Kneifel, S., S. Redl, E. Orlandi, U. Löhnert, M.P. Cadeddu, D.D. Turner, and M. Chen, 2014: Absorption Properties of Supercooled Liquid Water between 31 and 225 GHz: Evaluation of Absorption Models Using Ground-Based Observations. J. Appl. Meteor.

Climatol., 53, 1028–1045, https://doi.org/10.1175/JAMC-D-13-0214.1

Line 333: Very minor wording suggestion. Change "D14 puts emphasis at smaller particles" to "D14 emphasizes smaller particles".

Line 429: The authors use the subjective term "good" to describe GMI and RT simulation comparisons. Can a more quantitative or less subjective term be used here to describe the comparisons?

Line 453-454: I agree that ERA-Interim water vapour content is probably the likely culprit to cause clear sky biases. Clear sky RT results could also be influenced by the water vapour absorption model, but I'm not sure if that would cause the TB offset. What is the exact bias value under clear sky conditions? This would be useful information to convey. At lower microwave frequencies, ocean emissivity models can also be responsible for clear sky biases on the order of a few K. But surface effects are probably limited at some of these submillimetre channels, especially near water vapour absorption features.

Line 533: Capitalize Pacific Ocean?

Line 540: Change second sentence to "The biggest issues occur under clear-sky conditions, like due to [. . .]"

Lines 540-544: Regarding the deep convective cores with the lowest TB values, is it possible that graupel or hail aloft is not properly considered in the RT simulations? Would higher density particles be more appropriate under such conditions? This suggestion might increase the population of simulated extremely low TB values that already exceed the population of GMI TB values below about 170 K (Figure 10). But this suggestion might reduce the relative peak just below 150 K (Figure 10 bottom panels) by shifting the TB distribution to lower values and make the simulated TB distribution shaped more similarly to GMI observations. This is admittedly semi-educated speculation, but it might be another issue to highlight in the discussion section.

Related to the previous point, can the authors provide further useful analyses by partitioning their TB analyses using CloudSat-specific properties? The authors surmise, probably correctly, that certain TB regimes are related to deep convection, etc. CloudSat properties allow those assumptions to be ascertained without ambiguity and provide extremely valuable context. If the authors feel this type of analysis is beyond the current scope of the study, I would appreciate some justification. At a minimum, I encourage the authors to include language in the discussion section on possible ways to analyze the observations more deeply in follow-on studies. I envision multiple ways that CloudSat could inform the ICI simulations to better define TB simulation uncertainties for specific meteorological conditions based on cloud properties and related ambient conditions.

Lines 567-573: Passive infrared information would only provide cloud-top microphysical information, correct? The sensitivity to smaller particles would increase and further constrain the microphysical properties, but that information would only pertain to the cloud-top environment. Would the IR information advantage be related to the fact that if cloud-top microphysical properties could be better constrained, then microphysical evolution at each ensuing level below the cloud top is also better constrained?

---

## Author Comment (AC1) · 20 Nov 2019

Dear referee,

Please read the attached pdf for the authors replies.

Please also note the supplement to this comment:
https://www.atmos-meas-tech-discuss.net/amt-2019-293/amt-2019-293-AC1-supplement.pdf
* * *

---

## Author Comment (AC2) · 20 Nov 2019

**Response to reviewers**

Robin Ekelund, Patrick Eriksson and Simon Pfreundschuh

Department of Space, Earth and Environment
Chalmers University of Technology
Gothenburg
Sweden

November 20, 2019

First of all, the authors would like to give their thanks to the reviewers for their time and effort in reviewing our submitted paper. We also thank them for the overall positive assessment and their recommendations for publication after revision.

The comments are constructive and will improve the text in terms of clarity and scientific contribution; all of them will be addressed in some manner.

Below we respond to the critical comments. Comments on grammar and similar are omitted, but will be addressed in the revised manuscript.

**General response**

After considering the responses from the referees, we commit to perform the following main changes to the draft:

- The main concern from referee #1 is that the selection of frequencies is found to be limited. We've decided to complement the simulations with one additional channel, 166 GHz. However, for reasons given below, we argue that the other GMI and ICI frequencies are outside this article's scope.

- Referee # 2 found parts of the analysis too subjective and called for a more quantitative analysis of the output. We agree with this criticism, and will try to improve the manuscript in this respect. We will select and present some form of similarity measures for the presented TB distributions, which will aid the presentation and discussion of the forward simulations. It will hopefully also be useful for future potential studies adopting the method.

Also, after reading both of the referees' responses, we feel that the applicability, but also limitations, of the presented methodology could be made more clear.

**Responses to Anonymous Referee #1**

**General comments**

One of my main comments is why the GMI 89 and 166 GHz channels were not simulated. Trying to address the representativeness of ice particle models, excessive scattering at the lower microwave frequencies should be avoided and these channels could help. These channels would also give a very complete frequency-wise analysis of the most up to date database coupled with ARTS, which is extensively used. Is this outside the scope of this paper?

Similarly, with the 243 GHz channel in ICI.

A fair question. We realize that the choice of simulated frequencies with respect to GMI and ICI should be better motivated in the paper. We will try to motivate the channel selection to a higher degree in the revised manuscript.

To answer the question, more frequencies are of course always desired, but to some extent we were and are limited by computational resources and time. Including more frequencies would also make the results section too long. The intention of the article is to provide a demonstration of a method, not a full analysis of the scattering database. We selected the GMI and ICI channels that we deemed were the most interesting with respect atmospheric ice (those of higher frequencies).

Experience from test simulations done by us in the past, has shown that the 89 GHz channel (and other low frequencies) is more sensitive to rain and the melting layer, compared to ice hydrometeors. Hence, accurate modelling of this channel requires more rigorous treatment of rain and melting particles, which is outside the scope of this paper. Also, we found that 89 GHz provide very little additional information compared to CloudSat, likely due to the frequencies being so close together.

We agree that the 166 GHz is of interest in this case. Hence, we've decided to perform complementary simulations on this channel.

Regarding 243 GHz and the other ICI channels. Since there are currently no reference ICI data to compare to yet, there is little added benefit in simulating more ICI channels than the ones already covered. The ICI simulations are intended to show the potential of the presented method with respect to ICI and we argue that this has been already achieved. When ICI observations become available, a more complete frequency-coverage in a potential follow-up study is of course warranted.

**Specific comments**

Abstract: (L13) Could you please elaborate on what is meant by "a compensation effect between bulk extinction at passive frequencies and radar reflectivity."

The "compensation effect" is explained in the discussions using Fig. 7. Essentially, the spread between particle models are reduced when using the dBZ-mode, because the relationship between radar reflectivity and bulk extinction is relatively insensitive to assumed particle model (Fig. 7, middle panel).

We will either try to make the abstract text clearer or remove that specific part.

L168: For completeness perhaps it is worth including a comment about the suitability / experience when using the other scattering solvers in ARTS in the sub-millimetre range. Are there other publications where ARTS is used in that range?

We have previously performed tests and compared the different solvers implemented in ARTS (DISORT, Monte-Carlo, DOIT, RT4, etc.). Consistency within 1K comparing among the solvers were found. We selected DISORT for this study as it is fairly easy to use, stable, and insensitive to settings.

There are a several publications using ARTS in the sub-millimetre range which we can include the paper.

L193: What is the definition of aspect ratio?

The aspect ratio is in this case defined as the ratio between the minor (c) and major axes (a), i.e.. aspect ratio=c/a. This follows from "spheroids are oblate with an aspect ratio of 0.6'' since an oblate spheroid per definition has c<a and because the aspect ratio is less than 1.

We will clarify this in the revised manuscript.

L194: What sort of mixture was used? Ice in air, or air in ice?

Air in ice.

The air-fraction of this particle model varies widely. The smallest particles are completely composed of ice, while the largest are mostly composed of air (effectively being almost invisible at microwave wavelengths). Eriksson, et al. (2015) evaluated mixing-rules and found this setting to give the best agreement to DDA data.

We will clarify this in the revised manuscript.

L267: Also in the importance of consistency when making assumptions throughout the chain of simulations.

This intent of this comment is somewhat unclear to us. We agree with the statement, but do not understand the relevance here. We merely state that the purpose of the simulation mode in this specific study.

L285: Please discuss what is meant by "59 CloudSat orbits could be found and selected as references for the synthetic scenes". How coincident in time and space are you requiring CloudSat to overpass the GMI footprint.

To be clear, we don't look for collocated CloudSat and GMI measurements. We extract CloudSat and GMI observations from a set time frame (30 days in July, 2015) and geographic area (S20 to N20 and E170 to W130, in the Pacific ocean). CloudSat passes this area about two times a day, hence we get almost 60 scenes in total.

We selected this particular time frame in order to get as statistically close GMI and CloudSat observations as possible. CloudSat passes the equator in the Pacific around 13:30. We therefore looked for a time frame when the GPM passes the equator at roughly the same time in the selected region (bear in mind that GPM has a non-sun-synchronous orbit, meaning that the equatorial time drifts in longitude). The days where the orbits are the closest was found to be 15 July, hence we selected this month as the time frame.

From the CloudSat profiles, we derive synthetic scenes that are used as input to the forward calculations. The distributions of the simulated Tbs are then compared to the GMI observations.

We realize that this could be made more clear and will update the manuscript.

L289: By LWC do you mean cloud droplets? Please specify. I am assuming this because a few lines later you say you retrieve Rain Water content too.

Correct. We agree that the nomenclature using LWC and RWC is a bit confusing and we will address this in the text.

Note that LWC is not retrieved, but extracted from ERA-Interim (with some modifications explained in the text).

L393: lower TB-depressions (i.e., warmer simulated TBs). These are expected for the MH97 PSD since it's inclined to favour smaller particles?

Correct. We will elaborate on this in the text.

L400: Don't you mean the 186 GHz channel is closed to the centre of the water vapour line?

Correct. Typo.

At 328 GHz, the TB depression found for 334 GHz is masked by the stronger water vapour emission at higher altitudes. We will add this to the text.

We don't have a strong argument for doing this. Essentially, we thought it a good idea to visualize results from all PSDs for completeness, even though D14 only gets one panel. In essence, we've tried to highlight to most interesting features.

We understand that this reasoning may be considered ad hoc, even though the paper is of a demonstrative nature. Hence, in response to the first comment from referee #2 we will perform a more quantitative and systematic analysis of the results, as a complement to the visualizations (see our response to referee #2 for more details).

L458: I don't understand the comment about azimuthally oriented particles. If there were azimuthally oriented particles, shouldn't the TB depressions be actually larger hence even colder temperatures? You would increase retrieved IWC but you would be simulating colder too? This is the first concluding remark which needs revision, no horizontally aligned DDA shapes have ever been simulated.

What you're saying is what we are attempting to suggest.

To clarify, we try to explain why our simulated TB distributions are so cold. Lets assume the ice particle are horizontally aligned in reality. But if we assume random particle orientation, our model particles will scatters more weakly in the backward direction than in reality. Hence, we will overestimate IWC in our radar retrievals.

We then argue that the assumption on orientation does not impact the forward simulation significantly, because we assume 53° incidence (this was not explained clearly in the manuscript, we will fix this). The new article by Brath, et al. (2019) has simulated horizontally aligned DDA particles, and does indeed indicate that close to this angle the difference between horizontal and random orientation is not significant for intensity measurements (it is for polarization measurements, however). Hence, the passive forward simulations result in too high TB-depressions because we assume too high IWC.

On the other hand, Brath, et al. (2019) also indicate that at low Tbs, dense particles like graupel and hail dominate the scattering (i.e. particles with no orientation) because the polarization signal is suppressed. Referee #2 brings up the point incorrect treatment of denser particles could be problem as well.

Of course, all of this is currently only educated speculation, that may warrant a further study. We will clarify this in the manuscript.

L490: It would have been nice to evaluate this weak scatterer at the lower GMI frequencies.

As discussed previously, we will complement the simulations with 166 GHz. The 89 GHz channel we consider to be outside the scope of this study.

L520: I didn't catch the discussion made about the DARDAR spheroid.

The issue is that the DARDAR spheroid scattering data was produced by ourselves using T-matrix code, meaning that there might be inconsistencies compared to what was used in the DARDAR product. In order to validate our implementation of the particle, we tried to reproduce CloudSat reflectivities using the IWC and $N_0$*-fields from the DARDAR product. Due to the errors found, we acknowledge that there might indeed be some inconsistencies in our implementation of the particle.

We will try clarify this in the text.

L544: fairly insensitive at 190 GHz. Figure 11 (specially using F07T shows differently) Figure 10. It is interesting that the soft sphere with MH97 and dBZ-based mode, isn't worst than the sector snowflake.

We agree that it could be clarified that we here refer to the GMI simulations. Also, that the DARDAR spheroid seem to perform well for that particular case you mention is indeed interesting. Apparently we missed discussing this in the text and will address this.

**Responses to Anonymous Referee #2**

**General comments**

My only comment that could be considered as something more than a minor issue regards other possible options to provide further quantitative analyses that might be interesting to the community. I do not classify the suggestions as mandatory, but hopefully will spur the authors to find further creative ways to tabulate their comparisons to observations. Specific comments are outlined below.

We understand and agree with this concern. We will complement the results section with a more quantitative analysis using one or several measures for describing agreement or similarity between GMI and the RT simulations. It will include one or more tables that will provide a better overview of the results.

We have not yet decided upon exactly what type of measure will be presented. The difficulty lies in the fact that we are comparing distributions and not single measurements to each other. Hence, we can't derive statistics such RMSE, biases, etc. This is a limitation of the method that we will try to highlight more in the revised manuscript.

Example statistics could at the very least be the difference between mean simulated TB and mean observed GMI TB, but also some sort of measure on the deviation of the simulated TB distributions compared to GMI.

We also need to consider what type of information we want to convey and emphasize, which is not entirely obvious. For example, do we want to emphasize agreement at high altitude clouds or for deep convection? Such questions will influence the choice of measures to present.

Title: Since 664 GHz is considered in this study, maybe consider adding submillimetre to the title to better advertise the ICI applications? This suggestion is purely semantics, but this study extends beyond the wavelengths typically associated with microwave radiometry. Some remote sensing specialists designate submillimetre wavelengths as a distinct category occupying the space between microwave and infrared, while others may consider ICI-like frequencies as part of the microwave spectrum. Another option is to specifically include sensors like GPM and ICI in the title. This suggestion is not mandatory but is something the authors should consider to better advertise the novel ICI-related content.

We are aware of the distinction in how sub-millimetres are categorized (we tend to include sub-mm in MW) and agree that including sub-millimetres in the title may help to avoid confusion and promote the novelty of the content. As said, including GMI and/or ICI in the title could help in advertising the content. However, we are are also hesitant in doing so as it would detract from the focus on particle scattering data and the generality of the passive/active methodology.

Suggestion:

Using passive and active observations at microwave and sub-millimetre wavelengths to constrain ice particle models

**Specific comments**

Lines 276-278: A CloudSat-GMI coincident dataset exists, but as the authors mention, tropical coincidences are limited. Would it be worth highlighting how few coincident observations exist compared to higher latitudes? The Rysman et al. (2018) manuscript gives a quantitative analysis of global CloudSat-GMI coincidences for snowing observations (see Figure 2). At the very least, the Rysman et al. (2018) manuscript could be referenced to illustrate this point without the authors calculating their own statistics.

Rysman, J.-F.; Panegrossi, G.; Sanò, P.; Marra, A.C.; Dietrich, S.; Milani, L.; Kulie, M.S. SLALOM: An All-Surface Snow Water Path Retrieval Algorithm for the GPM Microwave Imager. Remote Sens. 2018, 10, 1278.

Good point. As suggested, we will highlight the scarcity of low-latitude observations and refer to the article and/or others.

Line 312: Please provide more information regarding gaseous and cloud liquid water absorption methodology used in the RT simulations. This information will allow other investigators to better replicate the study. Studies have also indicated RT variability using standard water vapour continuum and cloud liquid water absorption models (e.g., Turner et al 2009, Kneifel et al. 2014 and others), so knowing what absorption models were used is essential information.

D. D. Turner, M. P. Cadeddu, U. Lohnert, S. Crewell and A. M. Vogelmann, "Modifications to the Water Vapor Continuum in the Microwave Suggested by Ground-Based 150-GHz Observations," in IEEE Transactions on Geoscience and Remote Sensing, vol. 47, no. 10, pp. 3326-3337, Oct. 2009. doi: 10.1109/TGRS.2009.2022262

Kneifel, S., S. Redl, E. Orlandi, U. Löhnert, M.P. Cadeddu, D.D. Turner, and M. Chen, 2014: Absorption Properties of Supercooled Liquid Water between 31 and 225 GHz: Evaluation of Absorption Models Using Ground-Based Observations. J. Appl. Meteor. Climatol., 53, 1028–1045, https://doi.org/10.1175/JAMC-D-13-0214.1

Absorption coefficients are considered as follows: the gas absorption model PWR-98 (RosenKranz, 1998) is used for water vapour and oxygen, the standard profile by Liebe, et al. (1993) is used for nitrogen, and the liquid absorption model of Ellison, et al. (2007) is used for liquid clouds.

That there are uncertainties due to absorption models we don't argue against. While this is important when dealing with real observations, it's not of high concern for the conclusions made in this study.

The manuscript will be updated with this information.

Line 429: The authors use the subjective term "good" to describe GMI and RT simulation comparisons. Can a more quantitative or less subjective term be used here to describe the comparisons?

We understand this concern. As already mentioned, we will extend the manuscript with a more quantitate analysis. This will hopefully allow us to make the comparisons less subjective in general.

Line 453-454: I agree that ERA-Interim water vapour content is probably the likely culprit to cause clear sky biases. Clear sky RT results could also be influenced by the water vapour

absorption model, but I'm not sure if that would cause the TB offset. What is the exact bias value under clear sky conditions? This would be useful information to convey. At lower microwave frequencies, ocean emissivity models can also be responsible for clear sky biases on the order of a few K. But surface effects are probably limited at some of these submillimetre channels, especially near water vapour absorption features.

We agree that providing information on clear-sky TB biases could be useful. A quick check reveals that the difference (between simulated and GMI observations) in mean TB for clear sky conditions is about -5 and -3 K at 186 and 190 GHz. The increase in TB difference at 186 GHz (closer to the water vapour absorption line), provides a stronger indication to the water vapour hypothesis.

As an additional comment, Eriksson, et al. (2019) shows similar RT simulations with no significant clear-sky biases (compared to ATMS at 185.11 GHz), but covering all latitudes. Perhaps, the wet bias in ERA-Interim is a consequence of the prevalence of convection in the tropics.

As said, at these frequencies ocean emissivity should only play a minor role.

We will extend the paper to include this information and discussion.

Lines 540-544: Regarding the deep convective cores with the lowest TB values, is it possible that graupel or hail aloft is not properly considered in the RT simulations? Would higher density particles be more appropriate under such conditions? This suggestion might increase the population of simulated extremely low TB values that already exceed the population of GMI TB values below about 170 K (Figure 10). But this suggestion might reduce the relative peak just below 150 K (Figure 10 bottom panels) by shifting the TB distribution to lower values and make the simulated TB distribution shaped more similarly to GMI observations. This is admittedly semi-educated speculation, but it might be another issue to highlight in the discussion section.

This is an interesting point. To some extent, this is disproved by the TB distributions assuming the 8-col aggregate, which also shows a peak below 150 K. The particle model has a high effective density of 13.5 % (mass divided by volume of smallest circumsphere) which is fairly close to typical graupel densities, and is a strong scatterer as seen in Fig. 1.

However, as mentioned to referee #1, Brath, et al. (2019) does indeed indicates that hail and graupel (non-oriented scatterers) dominate at low Tbs since the polarization signal is reduced.

In order to test this once and for all, we might perform simulations with a higher density particle, if time and resources allow for it. We will then likely use the same particle model used for graupel in Brath, et al. (2019). However, for now we consider this as a lower priority compared to the other major changes decided to perform (given our argument on the 8-col aggregate). Regardless, we will bring up this point in the discussions.

Related to the previous point, can the authors provide further useful analyses by partitioning their TB analyses using CloudSat-specific properties? The authors surmise, probably correctly, that certain TB regimes are related to deep convection, etc. CloudSat properties allow those assumptions to be ascertained without ambiguity and provide extremely valuable context. If the authors feel this type of analysis is beyond the current scope of the study, I would appreciate some justification. At a minimum, I encourage the authors to include language in the discussion section on possible ways to analyze the observations more deeply in follow-on studies. I envision multiple ways that CloudSat could inform the ICI simulations to better define TB simulation uncertainties for specific meteorological conditions based on cloud properties and related ambient conditions.

We will try to meet the request made and try to incorporate CloudSat data in some way or another to enhance the analysis. Depending on the outcome of this, we will update the discussions as suggested.

As said, there are multiple ways to enhance the analysis using CloudSat data. As hinted, we could categorize the simulated TBs according to retrieved ice water path, cloud heights, etc. The cloud classification (2B-CLDCLASS) product is also a possibility.

To some extent, it might also be possible to give some recommendations on particle model or PSD for different conditions. However, the problem is that it is not straight-forward to evaluate this against the GMI observations, since in this study we can only compare TB distributions, not single observations.

Lines 567-573: Passive infrared information would only provide cloud-top microphysical information, correct? The sensitivity to smaller particles would increase and further constrain the microphysical properties, but that information would only pertain to the cloud-top environment. Would the IR information advantage be related to the fact that if cloud-top microphysical properties could be better constrained, then microphysical evolution at each ensuing level below the cloud top is also better constrained?

This is correct. We will elaborate the text to make this point clearer.

**References**

Brath, M., Ekelund, R., Eriksson, P., Lemke, O., and Buehler, S. A.: Microwave and submillimeter wave scattering of oriented ice particles, Atmos. Meas. Tech. Discuss., https://doi.org/10.5194/amt-2019-382, in review, 2019

Ellison, W.J. Permittivity of Pure Water, at Standard Atmospheric Pressure, over the Frequency Range 0–25THz and the Temperature Range 0–100C. J. Phys. Chem. Ref. Data 2007, 36, 1–18, [https://doi.org/10.1063/1.2360986]. Doi:10.1063/1.2360986.

Eriksson, P., Jamali, M., Mendrok, J., & Buehler, S. A. (2015). On the microwave optical properties of randomly oriented ice hydrometeors. Atmospheric Measurement Techniques, 8(5), 1913–1933. https://doi.org/10.5194/amt-8-1913-2015

Eriksson, P., Ekelund, R., Mendrok, J., Brath, M., Lemke, O., & Buehler, S. A. (2018). A general database of hydrometeor single scattering properties at microwave and sub-millimetre wavelengths. Earth Syst. Sci. Data, 10, 1301–1326. https://doi.org/10.5194/essd-10-1301-2018

Eriksson, P., Rydberg, B., Mattioli, V., Thoss, A., Accadia, C., Klein, U., and Buehler, S. A.: Towards an operational Ice Cloud Imager (ICI) retrieval product, Atmos. Meas. Tech. Discuss., https://doi.org/10.5194/amt-2019-312, in review, 2019.

Liebe, H.J.; Hufford, G.A.; Cotton, M.G., Eds. Propagation modeling of moist air and suspended water/ice particles at frequencies below 1000 GHz, 1993.

Rosenkranz, P.W. Water vapor microwave continuum absorption: A comparison of measurements and models. Radio Sci. 1998, 33, 919–928. doi:10.1029/98RS01182.

---

## Author Response (AR1)

**Response to reviewers**

Robin Ekelund, Patrick Eriksson and Simon Pfreundschuh

Department of Space, Earth and Environment
Chalmers University of Technology
Gothenburg
Sweden

November 20, 2019

Once again, the authors would like to give their thanks to the reviewers for their time and effort in reviewing our submitted paper. We also thank them for the overall positive assessment and their recommendations for publication after revision. Below we respond to the critical comments. The marked-up revised paper is appended at the end of this document.

**Major changes**

After considering the responses from the referees, we've performed the following main changes to the draft:

- The main concern from referee #1 is that the selection of frequencies is found to be limited. We've complemented the simulations with one additional channel, 166 GHz. However, for reasons given below, we argue that the other GMI and ICI frequencies are outside this article's scope.

- Referee # 2 found parts of the analysis too subjective and called for a more quantitative analysis of the output. We agree with this criticism, and have complemented the manuscript with a more quantitative analysis. This is found in the new section 4.2.4 "Overview of performance".

**Responses to Anonymous Referee #1**

**General comments**

One of my main comments is why the GMI 89 and 166 GHz channels were not simulated. Trying to address the representativeness of ice particle models, excessive scattering at the lower microwave frequencies should be avoided and these channels could help. These channels would also give a very complete frequency-wise analysis of the most up to date database coupled with ARTS, which is extensively used. Is this outside the scope of this paper?

Similarly, with the 243 GHz channel in ICI.

A fair question. We try to motivate the channel selection to a higher degree in the revised manuscript.

To answer the question, more frequencies are of course always desired, but to some extent we were and are limited by computational resources and time. Including more frequencies would also make the results section too long. The intention of the article is to provide a demonstration of a method, not a full analysis of the scattering database. We selected the GMI and ICI channels that we deemed were the most interesting with respect atmospheric ice (those of higher frequencies).

Experience from test simulations done by us in the past, has shown that the 89 GHz channel (and other low frequencies) is more sensitive to rain and the melting layer, compared to ice hydrometeors. Hence, accurate modelling of this channel requires more rigorous treatment of rain and melting particles, which is outside the scope of this paper. Also, we found that 89 GHz provide very little additional information compared to CloudSat, likely due to the frequencies being so close together.

We agree that the 166 GHz is of interest in this case. Hence, we've decided to perform complementary simulations on this channel.

Regarding 243 GHz and the other ICI channels. Since there are currently no reference ICI data to compare to yet, there is little added benefit in simulating more ICI channels than the ones already covered. The ICI simulations are intended to show the potential of the presented method with respect to ICI and we argue that this has been already achieved. When ICI observations become available, a more complete frequency-coverage in a potential follow-up study is of course warranted.

**Specific comments**

Abstract: (L13) Could you please elaborate on what is meant by "a compensation effect between bulk extinction at passive frequencies and radar reflectivity."

The "compensation effect" is explained in the discussions using Fig. 7. Essentially, the spread between particle models are reduced when using the dBZ-mode, because the relationship between radar reflectivity and bulk extinction is relatively insensitive to assumed particle model (Fig. 7, middle panel).

We decided to remove that specific part.

L168: For completeness perhaps it is worth including a comment about the suitability / experience when using the other scattering solvers in ARTS in the sub-millimetre range. Are there other publications where ARTS is used in that range?

We have previously performed tests and compared the different solvers implemented in ARTS (DISORT, Monte-Carlo, DOIT, RT4, etc.). Consistency within 1K comparing among the solvers were found. We selected DISORT for this study as it is fairly easy to use, stable, and insensitive to settings.

We've included several references on this.

L193: What is the definition of aspect ratio?

The aspect ratio is in this case defined as the ratio between the minor (c) and major axes (a), i.e.. aspect ratio=c/a. This follows from "spheroids are oblate with an aspect ratio of 0.6'' since an oblate spheroid per definition has c<a and because the aspect ratio is less than 1.

We have clarified this in the revised manuscript.

L194: What sort of mixture was used? Ice in air, or air in ice?

Air in ice.

The air-fraction of this particle model varies widely. The smallest particles are completely composed of ice, while the largest are mostly composed of air (effectively being almost invisible at microwave wavelengths). Eriksson, et al. (2015) evaluated mixing-rules and found this setting to give the best agreement to DDA data.

We have clarified this in the revised manuscript.

L267: Also in the importance of consistency when making assumptions throughout the chain of simulations.

This intent of this comment is somewhat unclear to us. We agree with the statement, but do not understand the relevance here. We merely state that the purpose of the simulation mode in this specific study.

L285: Please discuss what is meant by "59 CloudSat orbits could be found and selected as references for the synthetic scenes". How coincident in time and space are you requiring CloudSat to overpass the GMI footprint.

To be clear, we don't look for collocated CloudSat and GMI measurements. We extract CloudSat and GMI observations from a set time frame (30 days in July, 2015) and geographic area (S20 to N20 and E170 to W130, in the Pacific ocean). CloudSat passes this area about two times a day, hence we get almost 60 scenes in total.

We selected this particular time frame in order to get as statistically close GMI and CloudSat observations as possible. CloudSat passes the equator in the Pacific around 13:30. We therefore looked for a time frame when the GPM passes the equator at roughly the same time in the selected region (bear in mind that GPM has a non-sun-synchronous orbit, meaning that the equatorial time drifts in longitude). The days where the orbits are the closest was found to be 15 July, hence we selected this month as the time frame.

From the CloudSat profiles, we derive synthetic scenes that are used as input to the forward calculations. The distributions of the simulated Tbs are then compared to the GMI observations.

We have clarified this in the revised manuscript.

L289: By LWC do you mean cloud droplets? Please specify. I am assuming this because a few lines later you say you retrieve Rain Water content too.

We've changed LWC to CWC (cloud liquid water content).

Note that CWC is not retrieved, but extracted from ERA-Interim (with some modifications explained in the text).

L291: Please also refer the reader to section 3.2.

Done

L380: simulated TBs are indeed highly dependant upon assumed particle model (in the IWC retrieval).

Added parenthesis.

L393: lower TB-depressions (i.e., warmer simulated TBs). These are expected for the MH97 PSD since it's inclined to favour smaller particles?

Correct. We have clarified this in the revised manuscript.

L400: Don't you mean the 186 GHz channel is closed to the centre of the water vapour line?

Correct. Typo.

L402: 328 vs 334. Why especially different for the 9.2 degree?

At 328 GHz, the TB depression found for 334 GHz is masked by the stronger water vapour emission at higher altitudes.

We have clarified this in the revised manuscript.

L429: agreement to GMI (channels explored) is good.

Added parenthesis.

433: you mention that there are a few exceptions. It would be nice to include them in the text.

After revisiting the relevant figure, we've decided that there in essence are no significantly big differences at all (i.e., no remarkable exceptions). We've removed the "exception" from this sentence.

L438: Figure 11: why do you switch from MH97 to D14?

We don't have a strong argument for doing this. Essentially, we thought it a good idea to visualize results from all PSDs for completeness, even though D14 only gets one panel. In essence, we've tried to highlight to most interesting features.

L458: I don't understand the comment about azimuthally oriented particles. If there were azimuthally oriented particles, shouldn't the TB depressions be actually larger hence even colder temperatures? You would increase retrieved IWC but you would be simulating colder too? This is the first concluding remark which needs revision, no horizontally aligned DDA shapes have ever been simulated.

What you're saying is what we are attempting to suggest.

To clarify, we try to explain why our simulated TB distributions are so cold. Lets assume the ice particle are horizontally aligned in reality. But if we assume random particle orientation, our model particles will scatters more weakly in the backward direction than in reality. Hence, we will overestimate IWC in our radar retrievals.

We then argue that the assumption on orientation does not impact the forward simulation significantly, because we assume 53° incidence (this was not explained clearly in the manuscript, we will fix this). The new article by Brath, et al. (2019) has simulated horizontally aligned DDA particles, and does indeed indicate that close to this angle the difference between horizontal and random orientation is not significant for intensity measurements (it is for polarization measurements, however). Hence, the passive forward simulations result in too high TB-depressions because we assume too high IWC.

On the other hand, Brath, et al. (2019) also indicate that at low Tbs, dense particles like graupel and hail dominate the scattering (i.e. particles with no orientation) because the polarization signal is suppressed. Referee #2 brings up the point incorrect treatment of denser particles could be problem as well.

Of course, all of this is currently only educated speculation, that may warrant a further study. We have clarified this in the revised manuscript.

L490: It would have been nice to evaluate this weak scatterer at the lower GMI frequencies.

As discussed previously, we have complemented the simulations with 166 GHz. The 89 GHz channel we consider to be outside the scope of this study.

L520: I didn't catch the discussion made about the DARDAR spheroid.

The issue is that the DARDAR spheroid scattering data was produced by ourselves using T-matrix code, meaning that there might be inconsistencies compared to what was used in the DARDAR product. In order to validate our implementation of the particle, we tried to reproduce CloudSat reflectivities using the IWC and $N_0^*$-fields from the DARDAR product. Due to the errors found, we acknowledge that there might indeed be some inconsistencies in our implementation of the particle.

We have clarified this in the revised manuscript.

L544: fairly insensitive at 190 GHz. Figure 11 (specially using F07T shows differently) Figure 10. It is interesting that the soft sphere with MH97 and dBZ-based mode, isn't worst than the sector snowflake.

We have clarified this in the revised manuscript.

L464: simulations performed in this study (specially at 668.2 GHz)

Not sure how this fits into the text.

**Responses to Anonymous Referee #2**

**General comments**

My only comment that could be considered as something more than a minor issue regards other possible options to provide further quantitative analyses that might be interesting to the community. I do not classify the suggestions as mandatory, but hopefully will spur the authors to find further creative ways to tabulate their comparisons to observations. Specific comments are outlined below.

This question highlighted the difficulty in providing quantitative analysis using the method presented in this paper (i.e., not using collocated observations). We decided to calculate differences in mean simulated and GMI observed Tbs, only considering Tbs below 240 K. It was found that this metric gives a fairly good idea of the ability of given microphysical parametrization to reproduce the GMI distribution.

Analysis of all GM-simulations using this metric was added in the new section 4.2.4.

Title: Since 664 GHz is considered in this study, maybe consider adding submillimetre to the title to better advertise the ICI applications? This suggestion is purely semantics, but this study extends beyond the wavelengths typically associated with microwave radiometry. Some remote sensing specialists designate submillimetre wavelengths as a distinct category occupying the space between microwave and infrared, while others may consider ICI-like frequencies as part of the microwave spectrum. Another option is to specifically include sensors like GPM and ICI in the title. This suggestion is not mandatory but is something the authors should consider to better advertise the novel ICI-related content.

We are aware of the distinction in how sub-millimetres are categorized (we tend to include sub-mm in MW) and agree that including sub-millimetres in the title may help to avoid confusion and promote the novelty of the content. As said, including GMI and/or ICI in the title could help in advertising the content. However, we are are also hesitant in doing so as it would detract from the focus on particle scattering data and the generality of the passive/active methodology.

We changed the title to:

Using passive and active observations at microwave and sub-millimetre wavelengths to constrain ice particle models

**Specific comments**

Line 44: Should MLS, SMILES, and Odin-SMR acronyms be explicitly written?

Done

Lines 58-59: A reference or references might be beneficial to prove that soft spheroid models produce good results at single frequencies.

Added a reference (Galligani, et al. 2015).

Lines 276-278: A CloudSat-GMI coincident dataset exists, but as the authors mention, tropical coincidences are limited. Would it be worth highlighting how few coincident observations exist compared to higher latitudes? The Rysman et al. (2018) manuscript gives a quantitative analysis of global CloudSat-GMI coincidences for snowing observations (see Figure 2). At the very least, the Rysman et al. (2018) manuscript could be referenced to illustrate this point without the authors calculating their own statistics.

Rysman, J.-F.; Panegrossi, G.; Sanò, P.; Marra, A.C.; Dietrich, S.; Milani, L.; Kulie, M.S. SLALOM: An All-Surface Snow Water Path Retrieval Algorithm for the GPM Microwave Imager. Remote Sens. 2018, 10, 1278.

Good point. We added this and one other reference. Since both references illustrates the point well, we decided to not calculate our own statistic, however.

Line 312: Please provide more information regarding gaseous and cloud liquid water absorption methodology used in the RT simulations. This information will allow other investigators to better replicate the study. Studies have also indicated RT variability using standard water vapour continuum and cloud liquid water absorption models (e.g., Turner et al 2009, Kneifel et al. 2014 and others), so knowing what absorption models were used is essential information.

D. D. Turner, M. P. Cadeddu, U. Lohnert, S. Crewell and A. M. Vogelmann, "Modifications to the Water Vapor Continuum in the Microwave Suggested by Ground-Based 150-GHz Observations," in IEEE Transactions on Geoscience and Remote Sensing, vol. 47, no. 10, pp. 3326-3337, Oct. 2009. doi: 10.1109/TGRS.2009.2022262

Kneifel, S., S. Redl, E. Orlandi, U. Löhnert, M.P. Cadeddu, D.D. Turner, and M. Chen, 2014: Absorption Properties of Supercooled Liquid Water between 31 and 225 GHz: Evaluation of Absorption Models Using Ground-Based Observations. J. Appl. Meteor. Climatol., 53, 1028–1045, https://doi.org/10.1175/JAMC-D-13-0214.1

Absorption coefficients are considered as follows: the gas absorption model PWR-98 (RosenKranz, 1998) is used for water vapour and oxygen, the standard profile by Liebe, et al. (1993) is used for nitrogen, and the liquid absorption model of Ellison, et al. (2007) is used for liquid clouds.

The manuscript is updated with this information.

That there are uncertainties due to absorption models we don't argue against. While this is important when dealing with real observations, it's not of high concern for the conclusions made in this study.

Line 333: Very minor wording suggestion. Change "D14 puts emphasis at smaller particles" to "D14 emphasizes smaller particles".

Done.

Line 429: The authors use the subjective term "good" to describe GMI and RT simulation comparisons. Can a more quantitative or less subjective term be used here to describe the comparisons?

We changed the this sentence.

Line 453-454: I agree that ERA-Interim water vapour content is probably the likely culprit to cause clear sky biases. Clear sky RT results could also be influenced by the water vapour absorption model, but I'm not sure if that would cause the TB offset. What is the exact bias value under clear sky conditions? This would be useful information to convey. At lower microwave frequencies, ocean emissivity models can also be responsible for clear sky biases on the order of a few K. But surface effects are probably limited at some of these submillimetre channels, especially near water vapour absorption features.

A quick check reveals that the difference (between simulated and GMI observations) in mean TB for clear sky conditions is about -0.1, -4.7 and -2.6 K at 166, 186 and 190 GHz. The increase in TB difference at 186 GHz (closer to the water vapour absorption line), provides a stronger indication to the water vapour hypothesis.

We have added this information to the paper.

As an additional comment, Eriksson, et al. (2019) shows similar RT simulations with no significant clear-sky biases (compared to ATMS at 185.11 GHz), but covering all latitudes. Perhaps, the wet bias in ERA-Interim is a consequence of the prevalence of convection in the tropics.

Line 533: Capitalize Pacific Ocean?

Done

Line 540: Change second sentence to "The biggest issues occur under clear-sky conditions, like due to [. . .]"

Done

Lines 540-544: Regarding the deep convective cores with the lowest TB values, is it possible that graupel or hail aloft is not properly considered in the RT simulations? Would higher density particles be more appropriate under such conditions? This suggestion might increase the population of simulated extremely low TB values that already exceed the population of GMI TB values below about 170 K (Figure 10). But this suggestion might reduce the relative peak just below 150 K (Figure 10 bottom panels) by shifting the TB distribution to lower values and make the simulated TB distribution shaped more similarly to GMI observations. This is admittedly semi-educated speculation, but it might be another issue to highlight in the discussion section.

This is an interesting point. To some extent, this is disproved by the TB distributions assuming the 8-col aggregate, which also shows a peak below 150 K. The particle model has a high effective density of 13.5 % (mass divided by volume of smallest circumsphere) which is fairly close to typical graupel densities, and is a strong scatterer as seen in Fig. 1.

However, as mentioned to referee #1, Brath, et al. (2019) does indeed indicates that hail and graupel (non-oriented scatterers) dominate at low Tbs since the polarization signal is reduced.

We have extended the discussion in the manuscript regarding this.

We would have liked to perform additional simulations with a higher density particle, but didn't have the time to do so. Given our argument on the 8-col aggregate, this was low-prioritized.

Related to the previous point, can the authors provide further useful analyses by partitioning their TB analyses using CloudSat-specific properties? The authors surmise, probably correctly, that certain TB regimes are related to deep convection, etc. CloudSat properties allow those assumptions to be ascertained without ambiguity and provide extremely valuable context. If the authors feel this type of analysis is beyond the current scope of the study, I would appreciate some justification. At a minimum, I encourage the authors to include language in the discussion section on possible ways to analyze the observations more deeply in follow-on studies. I envision multiple ways that CloudSat could inform the ICI simulations to better define TB simulation uncertainties for specific meteorological conditions based on cloud properties and related ambient conditions.

Partitioning the simulations using the CloudSat classification product, we found that around 70% of the cases with Tbs below 200K were classified as deep convection. We have provided a comment on this.

We've added a small paragraph in the discussions on how CloudSat could be used to improve agreement of simulations to GMI and why we omitted this here (simply put, we found it difficult to do so without severely extending scope of the paper).

Lines 567-573: Passive infrared information would only provide cloud-top microphysical information, correct? The sensitivity to smaller particles would increase and further constrain the microphysical properties, but that information would only pertain to the cloud-top environment. Would the IR information advantage be related to the fact that if cloud-top microphysical properties could be better constrained, then microphysical evolution at each ensuing level below the cloud top is also better constrained?

This is correct. We've elaborated a bit on this in the text to make it more clear.

**Technical corrections / Minor issues**

L165: Suggesting changing the phrase "It is intended to be as general as possible. For instance, radiation is described using the full Stokes vector notation and in terms of usage it behaves as a scripting language" to "It is intended to describe radiation using the full Stokes vector notation in the most general manner possible, allowing a large amount of user input flexibility. In itself ARTS behaves as a scripting language on its own."

Done.

L184: I should read "A particle mixture consists of pristine crystals" without the "a".

Done.

L346: Please mention that the colored lines represent IWP from radar inversions

Done.

[revised manuscript text omitted]